

# How COVID-19 related policies reshaped organic aerosol source contributions in central London

Gang I. Chen[1]*, Anja H. Tremper[1], Max Priestman[1], Anna Font[2], and David C. Green[1,3]

[1]MRC Centre for Environment and Health, Environmental Research Group, Imperial College London, 86 Wood Lane, London, W12 0BZ, UK

[2]IMT Nord Europe, Europe, Institut Mines-Télécom, Univ. Lille, Centre for Education, Research and Innovation in Energy Environment (CERI EE), 59000 Lille, France

[3]HPRU in Environmental Exposures and Health, Imperial College London, 86 Wood Lane, London, W12 0BZ, UK

*Correspondence to: Gang I. Chen (gang.chen@imperial.ac.uk)



# Abstract

Particulate matter (PM) poses both health and climate risks. Understanding pollution sources is therefore crucial for effective mitigation. Positive Matrix Factorization (PMF) of Aerosol Chemical Speciation Monitor (ACSM) data is a powerful tool to quantify organic aerosol (OA) sources. A year-long study of ACSM data from London's Marylebone Road monitoring station during the COVID-19 pandemic provides insights into the impact of lockdown and the Eat Out To Help Out (EOTHO) scheme, which offered support to the hospitality during the pandemic, on PM composition and OA sources. Five OA sources were identified including hydrocarbon-like OA (HOA, traffic-related, 11% to OA), cooking OA (COA, 20%), biomass burning OA (BBOA, 12%), more-oxidized oxygenated OA (MO-OOA, 38%), and less-oxidized oxygenated OA (LO-OOA, 21%). Lockdown significantly reduced HOA (-52%), COA (-67%), and BBOA (-41%) compared to their pre-COVID levels, while EOTHO increased COA (+38%) significantly compared to the post-lockdown period. However, MO-OOA and LO-OOA were less affected, as these primarily originated from long-range transport. This research has highlighted the importance of commercial cooking as a significant source of OA (20%) and $PM_1$ (9%) in urban areas. The co-emission of BBOA with COA observed in Central London demonstrates a similar diurnal cycle and response to the EOTHO policy, indicating that cooking activities might be currently underestimated and contribute to urban BBOA. Therefore, more effort is required to quantify this source and develop targeted abatement policies to mitigate emissions as currently limited regulation is in force.





# 1 Introduction

Atmospheric particulate matter (PM) are tiny particles suspended in the air, which can not only impact the climate directly and indirectly (IPCC, 2021; Seinfeld et al., 2006), but also cause adverse health effects to human (Kelly and Fussell, 2012; World Health Organization, 2021). PM consist of various constituents, including inorganic species (metals, minerals, black carbon, nitrate, sulphate, etc.) and organic species (complex mixture of thousands of compounds). European Environment Agency has reported that 99% of urban population in Europe are still exposed to polluted air with annual $PM_{2.5}$ (PM with aerodynamic diameter smaller than 2.5 µm) concentrations exceeding the WHO air quality guideline, 5 µg/m$^3$ (Europe's air quality status 2024, 2024; World Health Organization, 2021). As the most health-relevant air pollutant, $PM_{2.5}$ has shown strong associations with cardiovascular and respiratory related mortalities and hospital admissions (Dominici et al., 2006; Wei et al., 2022, 2024). Several studies have demonstrated that different constituents/sources contribute to health effects differently with varying toxicities (Kelly and Fussell, 2012). Therefore, targeting the specific composition/sources of PM that are most health-relevant could be the most cost-effective way to mitigate its adverse health effects.

Source apportionment is a common but powerful approach to identifying and quantifying the emission sources and atmospheric constituents of PM based on measurements. As the sources of inorganic species (black carbon, ammonium, nitrate, chloride, sulphate, etc.) are relatively well-studied, most of the studies are focused on deconvoluting the sources of organic aerosol (OA), which contains thousands of compounds. Positive matrix factorization (PMF) is one of the receptor models that is widely utilized in the field to conduct source apportionment analysis. Typically, an Aerodyne aerosol mass spectrometer (AMS, Aerodyne Ltd., USA, Jayne et al., 2000) is used to measure the time series of both inorganic and organic species of non-refractory PM, in which,



organic mass spectra are used for PMF analysis. However, operating an AMS is labour-intense
and expensive. In contrast, the aerosol chemical speciation monitor (ACSM, Aerodyne, Ltd.,
Fröhlich et al., 2013; Ng et al., 2011) has been designed for long-term monitoring purposes with
less maintenance and lower capital cost, which has gained popularity across Europe (Chebaicheb
et al., 2024; Chen et al., 2022) and the U.S. (https://ascent.research.gatech.edu/). Chen et al. (2022)
demonstrated a robust protocol to conduct advanced PMF analysis on long-term ACSM datasets,
which delivers high-quality and consistent source apportionment results. This study follows this
standardized protocol to resolve the OA sources in London by implementing advanced PMF
techniques.
Coronavirus disease 19 (COVID-19) started to spread rapidly worldwide since the first case was
identified in Wuhan, China late in 2019. Many countries implemented measures to contain COVID
cases, which significantly restricted social and economic activities. In the UK, starting from the
end of Mar 26th, 2020, people were ordered to stay at home and all non-essential businesses were
closed, including pubs, cafes and restaurants. Non-essential shops were allowed to open on Jun
15th, and the first national lockdown came to an end Jun 23rd, 2020. However, pubs, restaurants,
and cafes were only allowed to open from July 4th, 2020. Subsequently, the Eat Out to Help Out
(EOTHO) Scheme was designed to help the hospitality industry; offering a 50% meal discount up
to a maximum of £10 and operated Monday to Wednesday during from Aug 3rd to Aug 31st, 2020;
https://www.gov.uk/guidance/get-a-discount-with-the-eat-out-to-help-out-scheme.
The UK recorded a 2.5% drop in Gross Domestic Product (GDP) in the first quarter of 2020, partly
as people reduced their own activity prior to the legally enforced lockdown measures introduced
on Mar 26th. This accelerated to a 19.8% fall in GDP in April to June 2020 and household spending
fell by over 20% over this period, the largest quarterly contraction on record, which was driven by



falls in spending on restaurants, hotels, transport, and recreation (GDP and events in history: how
the COVID-19 pandemic shocked the UK economy, 2024).
Some studies have investigated the lockdown impacts on chemical composition and sources of
PM, which mainly focused on cities in China (Hu et al., 2022; Tian et al., 2021; Xu et al., 2020),
a kerbside site in Toronto, Canada (Jeong et al., 2022), and an urban background site in Paris,
France (Petit et al., 2021). These studies all resolved primary sources including traffic related
emissions, biomass burning emissions from residential heating, cooking emissions (except Paris),
and secondary sources from PMF analysis on OA. Traffic and cooking emissions appeared to
decrease during the lockdown in all sites, while biomass burning predominately from residential
heating sources in Chinese cities increased as result of remote work and rather early lockdown
measures (Jan-Feb 2020) compared to France. Secondary organic aerosol (SOA) showed a more
complex phenomenon given its abundance in organic components and dynamic spatiotemporal
conditions. Overall, the lockdowns resulted in decreased SOA in both northwest cities in China
(Tian et al., 2021; Xu et al., 2020) and Paris (Petit et al., 2021) due to lower primary emissions,
and therefore fewer SOA formation products. However, Beijing experienced a large increase in
SOA concentrations due to increased fossil fuel and biomass emissions, long-range transport
influences as well as favourable meteorological conditions (high RH, low wind speed and low
boundary layer height) for SOA formation during the lockdown period (Hu et al., 2022). Therefore,
the lockdown effects on the SOA were dependent on the abundance of primary emissions, long-
range transported air masses, and meteorological conditions. To date, there are few studies that
investigate how COVID-related policies could have impacted PM chemical composition and
sources. Petit et al. (2021) and Gamelas et al., 2023) are only two studies in Europe. The unique
COVID-related policies in the UK provided a rare opportunity to investigate the impacts these



policies had on chemical composition and OA sources. To address these issues, we used highly
time resolved measurements from an air quality supersite located in the Central London from 2019
to 2020, and advanced source apportionment approaches to quantify the influence of the first
lockdown and EOTHO scheme on the PM composition and OA sources. This provides unique
insight into PM sources and composition in a global mega city.

## 110  2  Methodology

### 111  *2.1  Air quality monitoring supersite in central London*

The London Marylebone Road supersite (MY, 51.52 N, -0.15 E) is a kerbside monitoring site, one
meter away from a busy 6-lane road in central London. It is a well-established air quality supersite
that has consistently generated high-quality air pollution data since 1997 including mass
concentration of bulk $PM_1$, $PM_{2.5}$, and $PM_{10}$, as well as PM composition including black carbon,
heavy metals, nitrate ($NO_3$), sulphate ($SO_4$), ammonium ($NH_4$), OA, Chloride (Cl), etc. More
details of this site can be found at https://uk-air.defra.gov.uk/networks/site-info?site_id=MY1.

### 118  *2.2  Instrumentations*

Quadrupole ACSM (Q-ACSM, Aerodyne, Ltd., Ng et al. (2011)) provides 30-min mass loadings
of chemical species within non-refractory submicron aerosol (NR-$PM_1$), including $NH_4$, $NO_3$, $SO_4$,
Cl, and OA. Sampled particles are focused into a narrow beam using the aerodynamic lens and
impacted on a filament surface at 600 °C, where the NR-$PM_1$ is vaporised and ionised instantly by
an electron impact source (70eV). These ions are detected by the RGA quadrupole mass
spectroscopy to provide a mass spectrum of NR-$PM_1$ up to a mass-to-charge ratio (*m/z*) of 148 Th.
The mass concentration of different chemical species are calculated using the fragmentation table





126 developed by Allan et al. (2004), updated for Cl following suggestions provided by Tobler et al.

127 (2020), and a (Canagaratna et al., 2007; Matthew et al., 2008) composition-dependent collection

128 efficiency (CDCE) correction suggested by Middlebrook et al. (2012) by following the ACTRIS

129 standard operation procedure (https://www.actris-ecac.eu/pmc-non-refractory-organics-and-

130 inorganics). With co-located black carbon (BC) measurement using a $PM_{2.5}$ cyclone with AE33

131 (Aerosol Magee Scientific, Ltd.) and $PM_1$ measurements using FIDAS (Palas, GmbH), we

132 conducted the mass closure for fine particles measurements. The sum of NR-$PM_1$ and BC (in $PM_{2.5}$)

133 reproduces $PM_1$ concentrations well, with a slope of 1.13 and an $R^2$ of 0.73 (Fig. S1).

134 *2.3 Sampling periods and COVID-related policies*

135 $PM_1$ chemical composition from Aug 1st, 2019 to Oct 22nd, 2020, was analysed as this covered the

136 first lockdown period (Mar 26th–23 Jun 23rd, 2020) and the EOTHO Scheme (Mon-Wed during

137 from Aug 3rd to Aug 31st, 2020, Table 1). In order to isolate the seasonal effects on the PM chemical

138 composition and OA sources from the COVID-related policies, we further split the data based on

139 seasons (Table 1).




*Table 1 Dates of the COVID-related policies in London*

| COVID Policies | | Date |
|---|---|---|
| **Pre-Lockdown** | Summer | Aug 1$^{st}$–Aug 31$^{st}$, 2019 |
| | Fall | Sep 1$^{st}$–Nov 30$^{th}$, 2019 |
| | Winter | Dec 1$^{st}$, 2019–Feb 28$^{th}$, 2019 |
| | Spring | Mar 1$^{st}$–Mar 25$^{th}$, 2020 |
| **Lockdown** | Spring | Mar 26$^{th}$–May 31$^{st}$, 2020 |
| | Summer | Jun 1$^{st}$–Jun 23$^{rd}$, 2020 |
| **Post-Lockdown** | Pre-EOTHO | Jun 24$^{th}$–Aug 2$^{nd}$, 2020 |
| | EOTHO | Aug 3$^{rd}$–Aug 31$^{st}$, 2020 |
| | Post-EOTHO | Sep 1$^{st}$–Oct 22$^{nd}$, 2020 |

*2.4   Source apportionment*
Positive matrix factorization (PMF) has been widely deployed in source apportionment of PM
components including OA from ACSM/AMS datasets collected worldwide (Chebaicheb et al.,
2024; Chen et al., 2022; Jimenez et al., 2009; Ng et al., 2011b; Zhang et al., 2011). The PMF
algorithm on environmental monitoring data was initially introduced by Paatero and Tapper, (1994)
as follows:

$$x_{ij} = \sum_{k=1}^{p} g_{ik} \times f_{kj} + e_{ij} \tag{1}$$

where $x_{ij}$ is the measurement matrix (here, the time series of organic mass spectra from the ACSM
at $i^{th}$ time and $j^{th}$ *m/z*), $g_{ik}$ is the mass concentration at $i^{th}$ *time* in $k^{th}$ factor, $f_{kj}$ is the relative





intensity of $j^{th}$ *m/z* for $k^{th}$ factor, and $e_{ij}$ stands for the residuals for $j^{th}$ *m/z* at $i^{th}$ time, *p* is the number
of factors. The PMF model iteratively minimises the *Q* value using the least-squares algorithm as:

$$Q = \sum_{i=1}^{n} \sum_{j=1}^{m} \left(\frac{e_{ij}}{\sigma_{ij}}\right)^2 \qquad (2)$$

where *n* is the number of data points, *m* is the total number of *m/z*, and $\sigma_{ij}$ is the measurement
uncertainty estimated before the PMF analysis at $i^{th}$ time for $j^{th}$ *m/z*.
However, PMF suffers from rotational ambiguity (Paatero et al., 2002), which provides non-
unique solutions (i.e., similar Q value with different time series and factor profiles). These
solutions typically will not be equally environmentally reasonable, even with similar Q values.
The multilinear engine ME-2 (Paatero, 1999; Paatero and Hopke, 2009) is a robust approach to
reduce the rotational ambiguity and can direct PMF towards environmentally reasonable solutions
(both factor profiles and time series).
Here, PMF was implemented using the Source Finder v9.5.1.3 (Datalystica Ltd., Switzerland,
Canonaco et al. 2013) with the ME-2 solver. The latter imposes *a priori* information on the factor
solutions and/or time series. The *a*-value (ranging from 0 to 1) represents the upper limit of the
relative deviation for a factor profile ($f_j$) or time series ($g_i$) from the chosen *a priori* input profile
($F_j$) or time series ($G_i$) during the iterative least-square minimization (Equation 2), as shown in
Equations 3a and 3b (Canonaco et al., 2013):

$$f_j = F_j \pm a \cdot F_j \qquad (3a)$$

$$g_i = G_i \pm a \cdot G_i \qquad (3b)$$



PMF analysis is usually performed using the whole dataset, assuming that the OA source profiles
are static over the entire period, which can lead to high errors when it comes to long-term datasets
with non-negligible temporal variabilities of OA chemical fingerprints. Canonaco et al. (2015)
showed a considerable seasonal variability of oxygenated organic aerosol (OOA) factor profiles,
especially between winter and summer in a dataset in Switzerland. Parworth et al. (2015) first
introduced the concept of rolling PMF by shortening the analysis period to a smaller time window
(e.g., 14 days) and then rolling over the whole dataset with a certain step (i.e., 1 day). This
technique was further refined and implemented into SoFi by Canonaco et al. (2021), which allows
the PMF model to adapt the temporal variabilities of the source profiles (e.g., biogenic versus
biomass burning influences on OOA factors), which usually provides well-separated OA factors.
Bootstrapping (Efron, 1979) analysis will randomly select part of the PMF input matrix and
duplicates itself to recreate a matrix with the same dimension as the original PMF input matrix.
The statistical and rotational uncertainties of the PMF results will then be evaluated by bootstrap
and the random *a*-value approach with at least 50 repeats per rolling window (Canonaco et al.,
2021; Chen et al., 2021). The standardized protocol of rolling PMF as presented in Chen et al.
(2022) was used to ensure high-quality and comparable sources of OA were retrieved in London.
Specifically, PMF was first done on four different seasons as suggested in  Chen et al. (2022) to
determine the optimum number of factors. A total of 5 OA factors were identified: hydrocarbon-
like OA (HOA), cooking-like OA (COA), biomass burning OA (BBOA), more-oxidized OOA
(MO-OOA) and less-oxidized OOA (LO-OOA). In addition, site-specific factor profiles were
derived for HOA, COA, and BBOA through a seasonal bootstrap PMF analysis for winter (Dec,
Jan, and Feb) and used as constraints as suggested in Chen et al. (2022) and Via et al. (2022).
However, the MY site is surrounded by many restaurants with prevalent cooking emissions. Thus,



the chemical fingerprint for both HOA and COA might not be fully separated. Therefore, we
constrained the trend of $NO_x$ time series, BBOA and COA profiles from a previous winter
bootstrap solution collected in London North Kensington (2015-2018, Chen et al., 2022) to retrieve
environmentally reasonable results with five factors, so-called base case solution. Then, a
bootstrap resampling analysis with 100 iterations and five factors was conducted by constraining
the factor profiles of HOA, COA, and BBOA from the base case with random a-value from 0.1-
0.5 with step of 0.1. It results in stable factor profiles of these three primary sources as shown in
Figure S2, which shows good agreements with published reference profiles (Chen et al., 2022;
Crippa et al., 2013).
By constraining primary factor profiles of HOA, COA, BBOA in Figure S2 (averaged bootstrap
results) and two additional unconstrained factors with bootstrap resampling and the random a-
value option (0.1-0.5, step of 0.1), rolling PMF is conducted with a time window of 14 days and a
step of 1 day. A criteria list including selections based on both time series and factor profiles as
shown in Table S1 was applied as per Chen et al. (2022). With the help of t-test in temporal-based
criteria (1-3), we can minimize subjective judgements in determining the environmentally
reasonable results. Eventually, 3,166 runs (14.1%) of the PMF runs were selected to average as
the final results with 4.9 % unmodelled data points, which is comparable with other rolling PMF
analyses (Chen et al., 2022).





# 3 Results and Discussions

## *3.1 Chemical composition of submicron PM for different periods around the COVID-19 Lockdown*

The average $PM_1$ mass concentration at MY site was 11 µg/m$^3$ for the study period with 44% OA, 21% $NO_3$, 15% $SO_4$, 16% BC, 5% $NH_4$, and 0.6% Cl. The distribution of the chemical composition on $PM_1$ varied depending on the season and variation was associated with the lockdown and EOTHO policies (Figure 1). $PM_1$ increased by 34% in lockdown spring (Mar 26$^{th}$–May 31$^{st}$, 2020) compared to pre-lockdown spring (Mar 1$^{st}$–Mar 25$^{th}$, 2020), as well as $NO_3$ and $NH_4$, the later most likely originated from enhanced agricultural emissions in spring from the UK and wider continental Europe (Aksoyoglu et al., 2020). It was further confirmed, through back trajectory analysis, that elevated $PM_1$ events (Mar 25$^{th}$–Mar 28$^{th}$, Apr 8$^{th}$–Apr 10$^{th}$, and Apr 15$^{th}$–Apr 17$^{th}$), where the result of airmasses passing over northern continental Europe (Figure S3). $NO_3$ concentration reduced in summer 2019 and 2020 as expected compared to spring or fall seasons due to the volatility of $NH_4NO_3$, while $SO_4$ concentrations increased in summer due to enhanced photochemistry. During the lockdown in spring $SO_4$ concentrations remained high, which was associated with long-range transport.



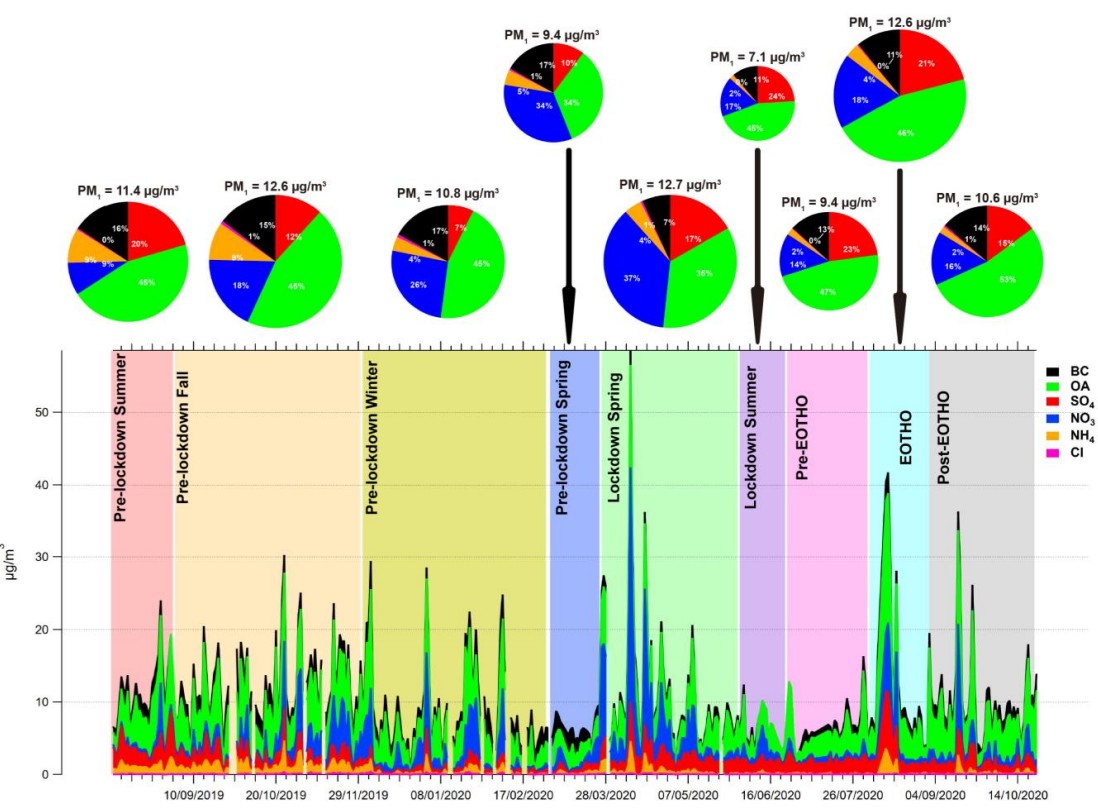

*Figure 1 Chemical compositions of PM₁ at MY from Aug 2019 to Oct 2020 (daily resolution) and averaged for the different periods as shown in Table 1.*

BC concentrations during the spring lockdown (Mar 26th–May 31st 2020) reduced from 1.59 to

0.87 µg/m³ (-45%) compared to the pre-lockdown level in spring (Mar 1st–Mar 25th), due to the

significant reduction in traffic during the first lockdown (Transport for London, 2020). It is worth

noting that the BC concentration had already reduced by 13% in pre-lockdown spring (Mar 1st–

Mar 25th) compared to the pre-lockdown winter. This is likely due to vehicle mileage reducing as

the UK government implemented travel restrictions and advised people to work from home on

Mar 16th, 2020 (Transport for London, 2020). BC increased to 1.24 µg/m³ (+57%) after the

lockdown and before the EOTHO (Jun 24th–Aug 2nd,2020, pre-EOTHO in Figure 1) as people

returned to work and travel. However, BC concentrations remained 31% lower than the pre-





lockdown summer (Aug 1$^{st}$–Aug 31$^{st}$, 2019) concentration of 1.8 µg/m$^3$, which suggests that the
traffic emissions reduced considerably as the fewer economic activities even after the ease of the
first lockdown (e.g., suggestions of hybrid working mode, restricted international travel, reduced
tourism, limited access to entertainments). BC also increased to 1.4 µg/m$^3$ (+10%) during the
EOTHO scheme (Aug 3$^{rd}$–Aug 31$^{st}$, 2020). This was not only because of increased traffic emission
during this period, but may also result from cooking activities (e,g, barbecuing or wood-fired
cooking styles) in central London. BC concentrations increased on Mon-Tue compared to post-
lockdown but before EOTHO (Jun 24$^{th}$–Aug 2$^{nd}$, 2020) (Figure S4).
*3.2   OA sources in Central London*
As mentioned above, the rolling PMF analysis resolved 5 factor solutions, including HOA, COA,
BBOA, MO-OOA, and LO-OOA as shown in Figure 2 and Figure 3. The left panel of Figure 2
shows the yearly averaged factor profiles of resolved PMF factors and total OOA calculated as the
sum of LO-OOA and MO-OOA. All factors show good agreements with previous studies in terms
of key *m/z*.
3.2.1   Time series of OA factors
Figure 2 shows both time series (30-min time resolution) and diurnal cycles for each OA factor.
The mean concentrations of HOA, COA, BBOA, MO-OOA, LO-OOA, and OOA were 0.50 ± 0.1
µg/m$^3$, 0.93 ± 0.14 µg/m$^3$, 0.55 ± 0.11 µg/m$^3$, 1.81 ± 0.41 µg/m$^3$, 1.00 ± 0.44 µg/m$^3$, and 2.80 ±
0.70 µg/m$^3$, respectively, and contributed to OA (PM$_1$) with the fractions of 11% (5% to PM$_1$), 20%
(9% to PM$_1$), 12% (5% to PM$_1$), 38% (17% to PM$_1$), 21% (9% to PM$_1$), and 59% (26% to PM$_1$).
The concentration of all OA factors shows strong time variations over the year as shown on the
left panel of the Figure 2. OA factors also showed strong seasonality besides the effects from



COVID-related policies (Figure 3). POA concentrations were generally lower in the warmer
seasons than in winter as lower temperature favours particle formation via condensation and
dilution and dispersion are reduced due to the lower boundary layer. The OOA factor
concentrations were larger the warmer seasons due to enhanced photochemistry at higher
temperature, stronger solar radiation and increased VOC emissions. The seasonality observed here
in central London agreed with the other sites across Europe (Chen et al., 2022).

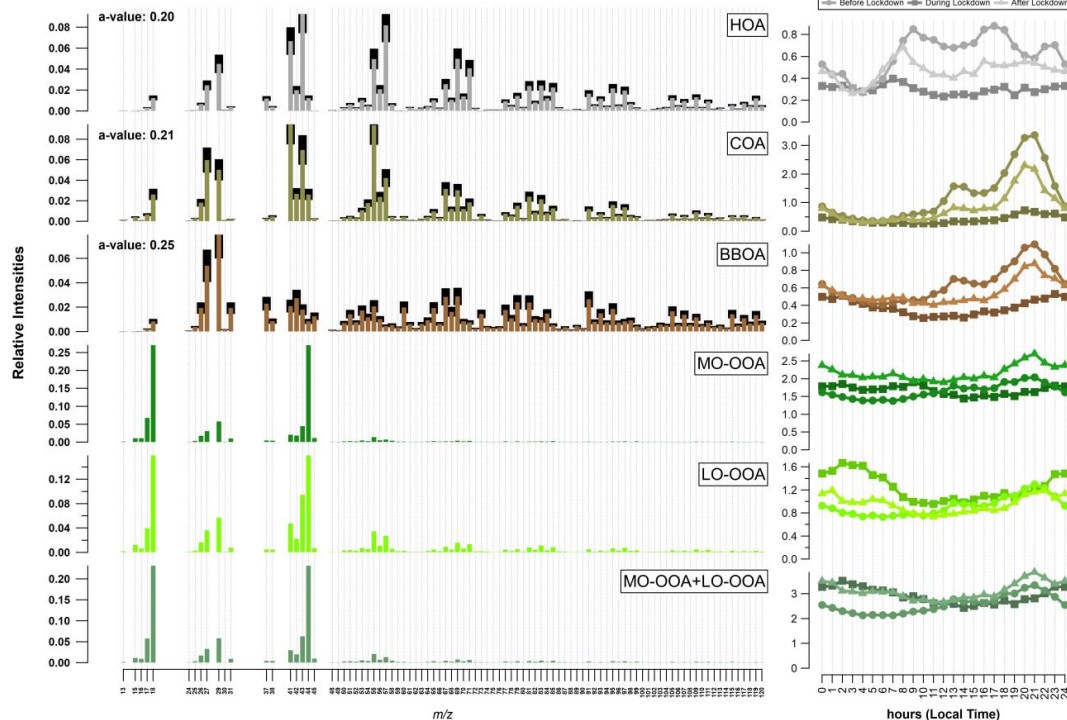


*Figure 2 Yearly averaged profiles (left) and diurnal cycles (right) of resolved factors from the rolling PMF analysis at the MY site.*
*Time is expressed in local time.*
3.2.2   Diurnal Cycles for OA factors
The right side of Figure 2 shows the diurnal cycles before, during, and after the lockdown. POA
factors showed distinct diurnal variations, in which HOA showed morning and evening rush hour
peaks, COA showed distinct lunchtime and evening peaks, and BBOA showed a similar pattern



as COA before and after the lockdown. This indicates that the part of what is measured as BBOA
in central London is most likely co-emitted from cooking activity, most likely from barbecuing
style restaurants in the area. Mohr et al. (2009) showed that meat-cooking can slightly elevate *m/z*
60, which is an important ion in the BBOA factor profile. OOA factors showed much less diurnal
variation compared to POA factors in all periods, this is in agreement with the other 22 European
sites reported in Chen et al. (2022). The MO-OOA showed a smaller diurnal variation compared
to LO-OOA.

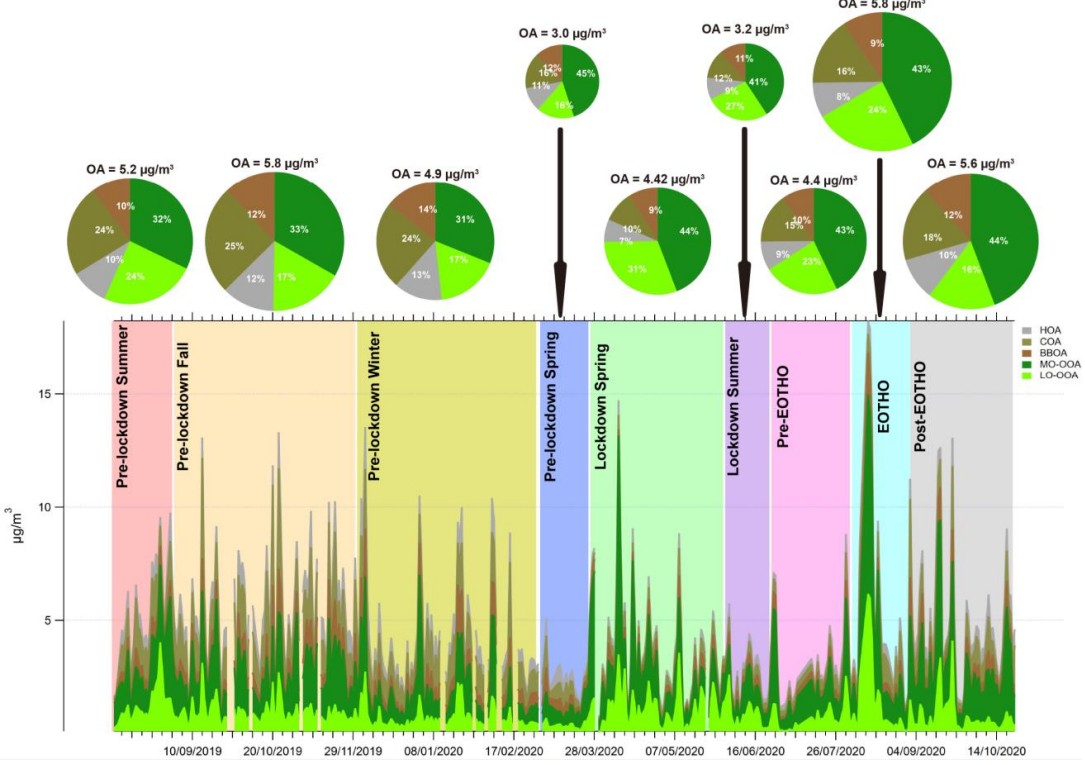

*Figure 3 Average mass concentrations for OA sources at MY during different periods from Aug 2019 to Oct 2020*
The diurnal variation of COA and BBOA during lockdown lost the distinctive lunch peak as shown
in the pre-lockdown; and the evening peak reduced its intensity (Figure 2). HOA retained distinct
morning and evening rush hour peaks but at lower mass concentrations during lockdown (Figure





2). After the first lockdown, the distinct lunch and evening peaks in diurnal patterns of COA and
BBOA reappeared as the open-up of nearby restaurants. The morning and evening rush hour peaks
for HOA enhanced considerably as the ease of the travel restrictions after the first lockdown.
However, POA concentrations did not reach pre-COVID levels. This is likely due to widespread
hybrid working and the remaining oversea travel restrictions supressing tourism, which reduced
traffic activity and restaurants visits. Conversely, OOA concentrations were slightly higher than
pre-lockdown levels. These were related to long-range transport, with relatively high mass
concentrations of MO-OOA and LO-OOA during the lockdown (Figure S3).

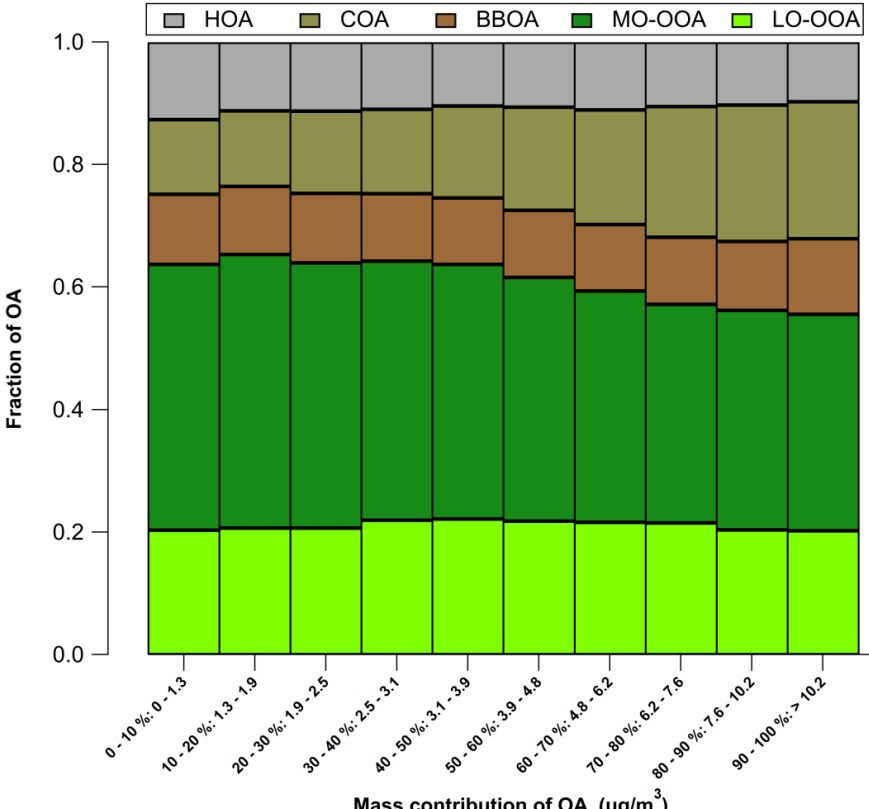


*Figure 4 Contributions to total OA from the different identified OA sources at different OA concentrations. Total OA*
*concentrations were split in 10 equally distributed bins.*



As shown in Figure *4* The contribution to total OA concentrations from HOA, BBOA, and LO-
OOA was consistent at different OA concentrations. However, the contribution of COA increased
as total OA concentrations increased (Figure *4*), This suggests that cooking emissions in Central
London are responsible for elevated OA concentrations.
### 3.2.3    Impact of lockdown on OA Concentrations
OA concentration decreased by 34% in pre-lockdown spring compared to pre-lockdown winter
(Dec 1st, 2019–Feb 28th, 2020) due to seasonality and the impact of lockdown. OOA concentrations
were relatively unaffected with some variability before, during, and after the lockdown. due to
long-range transportation of airmasses from the continental Europe as observed for $NH_4$, $NO_3$, and
$SO_4$. Primary emissions were significantly lower due to reduced vehicle mileage and other
economic activity before the official lockdown measure came into force (Figure 3 and Figure 2
(a)) as suggested by the 1st quarter drop in GDP. Atmospheric components related to vehicles
(HOA and BC) decreased by 50% and 13% respectively, in early March 2020. COA and BBOA
decreased by 60% and 47% respectively. COA, due to fewer restaurant activity, BBOA likely
reduced partly due to the warmer weather requiring less domestic space heating, and also due to
reduced commercial cooking using charcoal and wood.
Compared to the pre-lockdown spring, HOA and COA in the lockdown spring decreased by 8%
and 11%, respectively, while BBOA increased marginally by 5% (from 0.37 to 0.39 μg/m³) (Figure
3). MO-OOA and LO-OOA increased by 43% and 169%, respectively due to long-range
transportation of airmasses from continental Europe and increased photochemistry compared to
the first 25 days in Mar 2020. This was accompanied by increased $SO_4$ (+119%), $NH_4$ (+16%) and





$NO_3$ (+46%), despite the higher temperature could favour partitioning these species into the gas
phase.
In June 2020, still in lockdown (Jun 1st– Jun 23rd, 2020), POA showed further but marginal
decreases (-4%, -8%, and -10% for HOA, COA, and BBOA, respectively, Figure 3) compared to
the lockdown spring as the enhanced photochemistry leads to increased formation of OOA from
the POA. However, the overall mass concentration of MO-OOA and LO-OOA decreased
significantly by 34%, and 37%, respectively as the result of fewer long-range transported airmasses.
During pre-EOTHO (Jun 24th–Aug 2nd, 2020), HOA, COA, and BBOA all showed considerably
increases of 34%, 69%, and 25%, respectively when compared to lockdown summer period. In
which, MO-OOA and LO-OOA also increased by 45% and 18%, respectively as the results of
long-range transported airmasses from continental Europe, enhanced biogenic emissions and
photochemistry. The POA concentrations were much lower when compared to summer 2019 (Aug
1st–Aug 31st, 2019) as travel and economic activities did not return to pre-COVID levels.
Specifically, reduced vehicle mileage resulted in lower HOA (-22%), BC (-31%), COA (-46%)
due to the reduced commercial cooking activity. As BBOA is co-emitted with COA during of
cooking activities, BBOA also decreased slightly from 0.53 to 0.44 µg/m$^3$ (-17%, Figure 3).
3.2.4    Eat out to help out (EOTHO)
During EOTHO (Aug 3rd–Aug 31st, 2020), MO-OOA and LO-OOA increased by 31% and 35%
respectively compared to post-lockdown concentrations before EOTHO and correlated with
increased $NO_3$ and $SO_4$ concentrations. This was due to long-ranged transported airmasses and
enhanced photochemistry as well as the photooxidation of POAs.




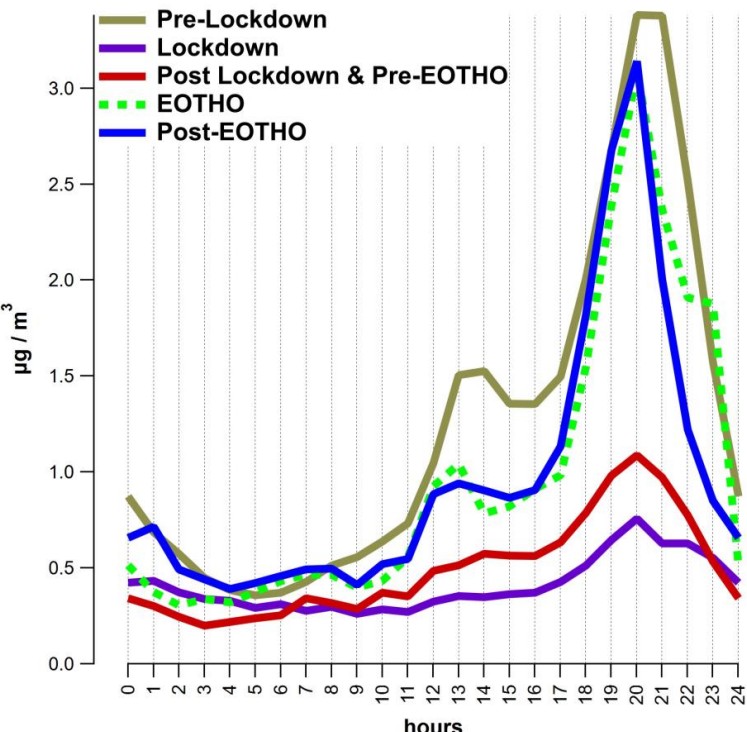

*Figure 5 . COA diurnal plots at different periods in relation with COVID-related policies*

However, EOTHO policy had a significant impact on all POA factors. In particular, the COA

concentration increased by 38% compared to the post-lockdown period from Jun 24th to Aug 2nd,

2020 (Figure 5). HOA and BBOA concentrations also increased by 22% and 23%, respectively,

which suggested the human activities resulting in these emissions recovered slowly after the

lockdown. COA was significantly higher due to EOTHO, however, it did not reach pre-COVID

concentrations (Figure 5). After the EOTHO policy (Sep 1st–Oct 22nd, 2020), COA concentrations

increased by 10% (Figure 5). This may have partially been due to lower temperatures, reduced

dispersion and photochemistry in Autumn.

EOTHO only operated from Mon to Wed, and this was clear in the diurnal plots (Figure 6) and

Figure S6 with larger COA concentrations Mon to Wed, in contrast with larger concentrations over

the weekend (Fri to Sun) before EOTHO (Jun 24th–Aug 2nd, 2020). Interestingly, even after the





EOTHO policy ceased, COA levels remained elevated on Mon and Tue but a much higher level
during the weekend was observed. This suggests that EOTHO had an influence on the consumer
behaviour even after the lockdown. It is also worth noting that the high concentrations of COA
and BBOA (Figure S5) on Monday night were caused by the last day of EOTHO policy coinciding
with a UK public holiday on Aug 31$^{st}$.

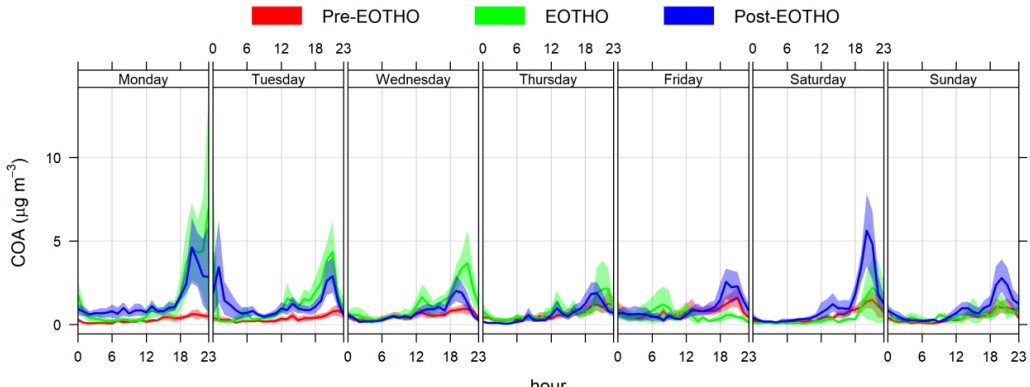


*Figure 6 The diurnal cycles for each day of the week in COA concentrations before, during, and after the Eat Out To Help Out*
*(EOTHO) policy in post-lockdown period (Jun 24$^{th}$–Oct 22$^{nd}$, 2020)*

## 4   Conclusion

This study demonstrates the importance of source apportionment studies to better understand how
national and local government policies can impact the PM mixture, and how these effects can be
differentiated from the influences of meteorology and large-scale atmospheric processes. PM
concentrations increased at the beginning of the lockdown (Mar–Apr 2020), coinciding with
reduced economic activities, however by examining the source apportionment (and inorganic PM
composition) the impact of lockdown policies on primary emissions could be quantified. COVID-
related policies were found to have profound but largely unintended impacts on air quality. The
first lockdown significantly reduced POA sources: including HOA by 52%, COA by 67%, and



BBOA by 41%. While all these components reduced dramatically during the lockdown, they only
gradually increased again and did not reach pre-COVID levels during the duration of this study.
Most significantly, while the Eat Out To Help Out (EOTHO) policy was effective in helping the
hospitality industry to recover from economic losses during the lockdown, it had unintended
impacts on air quality as cooking emissions increased. Clearly detecting this change confirms the
presence of COA (20% to OA) as an important source of OA in London, and other cities, and the
importance of commercial cooking as a source. Also of note was the impact that EOTHO had on
BBOA concentrations, which increased by 23% while this policy was in place. This establishes a
clear link between commercial cooking activity and BBOA measured in cities due to the use of
charcoal and wood as cooking fuels, as well as potentially emissions from cooking ingredients.
Cooking may therefore be underestimated as a source if COA concentrations are considered in
isolation, and BBOA is only associated with other sources of solid fuel burning. This emphasises
the need to develop policies and technical solutions to mitigate commercial cooking emissions in
the urban environment, especially as there are limited regulations on this industry in terms of air
pollution. It also demonstrated the importance in continuous monitoring with subsequent source
apportionment analysis to better understand the influence of government policies to improve air
quality more effectively.



# Code/Data availability

Rolling PMF analyses is run using SoFi Pro from Datalystica (https://datalystica.com/sofi-pro/, Datalystica, 2024) under Igor Pro 9 platform from WaveMetrics® (https://www.wavemetrics.com/, WaveMetrics, 2024) and they are both available for purchase. Raw data/results from the study are available upon request to the corresponding author Gang I. Chen (gang.chen@imperial.ac.uk).

# Author contribution

**Gang I. Chen:** Writing – review & editing, Writing – original draft, Visualization, Validation, Project administration, Methodology, Investigation, Funding acquisition, Formal analysis, Data curation, Conceptualization. **Anja H. Tremper:** Writing – review & editing, Methodology, Formal analysis, Data curation. **Max Priestman:** Methodology, Formal analysis, Data curation. **Anna Font:** Writing – review & editing, Methodology, Formal analysis, Data curation. **David C. Green:** Writing – review & editing, Supervision, Project administration, Methodology, Resources, Funding acquisition, Conceptualization.

# Competing interests

The authors declare that they have no known competing financial interests or personal relationships that could have appeared to influence the work reported in this paper.

# Acknowledgement

This study is part funded by the National Institute for Health Research (NIHR) Health Protection Research Unit in Environmental Exposures and Health, a partnership between UK Health Security



Agency (UKHSA) and Imperial College London. The views expressed are those of the authors
and not necessarily those of the NIHR, UKHSA or the Department of Health and Social Care. This
study was also supported by NERC Awards (NE/1007806/1) and (NE/T001909/2).

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
