# Peer review of "How COVID-19 related policies reshaped organic aerosol"

_EGUsphere, 2024_

## Author Comment (AC1)

**Response of the reviewers' comments on " *How COVID-19 related policies reshaped organic aerosol source contributions in central London"* by Gang I. Chen et al.**

*We thank all the constructive comments from two reviewers. The following texts are the response to the reviewers with normal italic font is original review comments*, *green font for authors' responses,* and the *blue italic font for changes in the revised version.*

*Review comments*

*Authors' response*

*Texts from the revised version*

**Reviewer 1**

*Chen et al. investigated the effect of COVID-19-related policies on aerosol composition in London, particularly how the source apportionment of organic aerosol varied before and after the implementation of different policies (lockdown and Eat Out to Help Out (EOTHO)). They found that the lockdown due to COVID-19 substantially reduced levels of primary organic aerosol (POA = hydrogen-like + cooking + biomass burning OAs (HOA + COA + BBOA)), lowering them by approximately 50% compared to pre-COVID levels. In contrast, oxygenated organic aerosols (OOA = less-oxidized OOA (LO-OOA) + more-oxidized OOA (MO-OOA)) were not as significantly affected. However, when EOTHO was introduced, POA levels— especially COA—increased, highlighting cooking activities as an important source of urban air pollution.*

*This study is a compelling example of why policymakers should consider unintended consequences when implementing policies. However, several aspects need improvement before publication. My primary recommendation is for the authors to reorganize the results and discussion sections to create a smoother flow that readers can easily follow and ensure the content aligns with the title. I suggest restructuring sections to first introduce the regional characteristics, then clearly describe the similarities and differences in OA characteristics during the pre-COVID, COVID, and EOTHO periods, while avoiding a time series-based presentation of the results.*

*That said, the authors present an excellent dataset that clearly illustrates changes in the characteristics of OA across the pre-COVID, COVID, and EOTHO periods. Accordingly, I suggest a major revision before publication in ACP. Additional comments are listed below.*

We appreciate the positive feedback from the reviewer, and we agree that this study has strong policy implications on cooking emissions in the urban environment. We

also agree with the reviewer regarding the restructuring of the manuscript and have implemented this in the corrected manuscript. The response to the additional comments is provided below.

*Major comments:*

*The ME-2 solution selection process should be described in greater detail. I expect to see more information in the SI on how the current solution differs from those obtained using a lower or higher number of factors. Additionally, it would be beneficial for the authors to explain how varying the a-value influences the source apportionment solutions. Lastly, I noticed that the a-value in Figure 2 is presented in a 0.## format, whereas the measurement section states that the a-value was adjusted in increments of 0.1. The authors should provide a comprehensive description of the source apportionment analysis, including the aspects mentioned above.*

Determining the number of factors is done during the seasonal PMF step. Doing rolling PMF with different number of factors is not feasible as it takes substantial time to do so. To clarify our selections of the number of factors, we've added two sentences as below in the main text in section 2.4 (line159-162 in the clean version):

"Adding an additional factor resulted in split of COA factor, decreasing it to four factors caused mixing between the MO-OOA and COA factors. Therefore, 5 factor-solution was determined across the whole year."

The sensitivity analysis of different a-values was initially conducted during the pre-test of seasonal PMF. The best solution was chosen as the base case solution for further bootstrap analysis based on factor profiles, time series, diurnal cycles, etc. Inappropriate a-values are typically easy to identify as the factor profiles/diurnal cycles will become mixed. In addition, the bootstrap analysis of the seasonal PMF enabled random a-value approach (Canonaco et al., 2021; Chen et al., 2021) by using the constraints retrieved from base case solution, which means the a-values for each constraint is allowed to vary from 0.1 to 0.5 with a step of 0.1 to evaluate the rotational uncertainties and temporal variabilities of PMF with 100 iterations (Chen et al., 2021), it has covered all the possible combinations of different a-values. Using the criteria-list mentioned in Table S1, we are able to select "good" PMF runs automatically with minimized subjective judgements. Similarly, the rolling PMF applies bootstrap using the random a-value approach from 0.1 to 0.5 with a step of 0.1 for HOA, COA, and BBOA with 50 iterations per rolling window (i.e., 14 days). The a-value showed in Figure 2 is the averaged a-values with 2 decimal places for selected PMF runs for all the windows throughout the year. That's why the averaged a-values have two digits. To avoid confusion, line 174-183 (clean version) has been revised as follows:

"*Rolling PMF was conducted with a time window of 14 days and a step of 1 day by constraining primary factor profiles of HOA, COA, BBOA in Figure S2 (averaged bootstrap results) and two additional unconstrained factors with bootstrap resampling and the random a-value option (0.1-0.5, step of 0.1, 50 iterations/window). A criteria list including selections based on both time series and factor profiles as shown in Table S1 was applied as per Chen et al. (2022). With the help of t-test in temporal-based criteria (1-3), we can minimize subjective judgements in determining the environmentally reasonable results. Eventually, 3,166 runs (14.1%) of the PMF runs were selected across different rolling windows across the whole year to average as the final results (utilized a-values were averaged to two decimal places) with 4.9 %*

*unmodelled data points, which is comparable with other rolling PMF analyses (Chen et al., 2022).*"

As the detailed methodology have been described in (Chen et al., 2022) with step-by-step instructions, and the scope of this study is to focus on the OA characteristics under different policies. Therefore, more detailed descriptions of the pre-test PMF is not provided.

1. I expect to see a more detailed description in Section 3.1, not just the changes in mass concentration. For example, how did the fraction of non-refractory species vary by season or implementation of policies? How was this particular period different from the past PM variation from literature? Also, when comparing the PM characteristics between pre-lockdown vs. post-lockdown, I think it should be more clearly compared by season.

   The scope of this study is to highlight the influences of COVID-related policies on source apportionment results of organic aerosols. In general, the COVID-related policy did not have significant influences on PM species (except for BC) and the seasonal variabilities of PM species was quite similar to other European sites as described in Bressi et al. (2021). Therefore, the variabilities of PM species over different periods were briefly discussed except for the BC, which is discussed extensively.

2. I recommend the authors revisit the discussion regarding OA factors because I see multiple points that are difficult to agree with the authors' perspectives. For example, it is hard for me to recognize a strong seasonality in Figure 3 while only the effects of COVID-related policies are noticeable. In addition, I think there is a small lunchtime peak in the diurnal of COA if you zoom in. Additionally, the authors should add more detailed discussion by comparing with previous studies that were conducted in other major cities around the world. I added further details in the specific comments below.

   We agree that Figure 3 is difficult to see the seasonality of OA sources. This is common in the UK, which is influenced by changeable weather systems, and seasonality of OA sources in another London site (urban background) have been discussed extensively in (Chen et al., 2022). Both datasets showed a similar seasonality for all the sources.

   The statement about COA diurnal variation during lockdown has been amended to avoid confusion, as indeed it still showed small lunchtime peak, likely due to takeout activity of some restaurants that were still active as well as potential increases for residential cooking activities at that time. Line 291-294 (clean version).

   *"The diurnal variation of COA and BBOA during lockdown showed much less intensity overall but the distinctive lunchtime peak remained as the pre-lockdown; and the evening peak reduced its intensity (**Error! Reference source not found.**). This is because the takeout activities of some restaurants were still active as well as the potential increases for residential cooking activities during lockdown."*

   As mentioned in the introduction, there are not many OA source apportionment studies during the lockdown periods around the world. Majority of studies were in China, where the lockdown time was different from the UK and the rest of the world. Also, most of studies only showed the effects of lockdown, which demonstrated the decrease

of primary organic aerosol (POA) sources. However, the unique EOTHO policy in the UK cannot be compared with other cities.

3. In some sections, it was hard to follow the authors' discussion due to the lack of results or references that support authors' opinions. A couple of examples can be found in Section 3.2. If the authors can provide any results (or references) that show changes in temperature, GDP changes, travel, economic activities, vehicle mileage, etc., it would be much more supportive of the authors' discussion in Section 3.2.3 and 3.2.4, where focused on policy-related effects on OA.

Citations have been provided to support authors' opinions

4. I recommend that the authors add a separate section at the beginning of Section 3.2 to describe the typical OA characteristics in the region. This would help guide readers and establish a clearer baseline for understanding the impact of the policies on OA characteristics. Additionally, I suggest restructuring the paper as follows: (1) general OA characteristics of the region, (2) pre-lockdown, (3) lockdown, and (4) EOTHO. From my perspective, the current structure, which includes an overall time series and diurnal section, weakens the paper's main argument and leads to repetitive statements (e.g., long-range transport, seasonality of photochemistry).

We appreciate reviewer's comment, and we have revised the section 3.2 according as shown in the clean version.

**Specific comments:**

1. Line 119: Is this ACSM deployed with a standard vaporizer? Or a capture vaporizer?

It's a standard vaporizer. Revised as follows:

*"Quadrupole ACSM (Q-ACSM, Aerodyne, Ltd., Ng et al. (2011)) with a standard vaporizer provides 30-min mass loadings of chemical species within non-refractory submicron aerosol (NR-PM$_1$), including NH$_4$, NO$_3$, SO$_4$, Cl, and OA.*

2. Line 218: What kind of activities are related to elevated PM1 events? Did airmass originate from northern continental Europe and also be related to agricultural activities? Please specify."

Agreed. Additional sentences have been added as follow:

*"PM$_1$ increased by 95% in lockdown spring (Mar 26$^{th}$–May 31$^{st}$, 2020) compared to pre-lockdown spring (Mar 1$^{st}$–Mar 25$^{th}$, 2020). Specifically, Org, SO$_4$, NO$_3$, NH$_4$, and Cl all increased by 87%, 211%, 73%, 237%, and 132%, respectively. Except for BC, which decreased by 52%. This is due to the polluted airmass originating from mainland Europe and the enhanced agricultural emissions in spring from the UK and wider continental Europe (Aksoyoglu et al., 2020)."*

3. Line 220: Would only volatility be related to lower NO3 concentration in summer? How about NOx/NH3 emission or other factors that can affect the formation of inorganic NO3?

The main reason is its volatility but there are also less agricultural emissions in summer compared with spring and fall seasons. Revised as follows:

*"NO$_3$ concentration reduced in summer 2019 and 2020 as expected compared to spring or fall seasons due to the volatility of NH$_4$NO$_3$ and lower agricultural emissions, while SO$_4$ concentrations increased in summer due to enhanced photochemistry."*

4. Line 221: Please add a citation and describe briefly how SO4 is formed via photochemistry. Also, how did SO2 change throughout the measurement period?

   Two citations (Bressi et al., 2021; Chen et al., 2022) have been added where it showed same seasonal variations of enhanced SO$_4$ in summer due to more intense photochemistry across Europe. Also, Figure S7 (also shown in point 5) was added, which demonstrated the SO4 was originated from Eastern Europe.

   SO$_2$ concentration decreased significantly with a step change during lockdown and remained at a low level until the end of the measurement period as shown below with a rapid drop (20%) in cargo volumes in the UK in the 2$^{nd}$ quarter of 2020 (Simmonds & George Finch, 2021). This may be the reason that SO$_4$ in lockdown summer remained at a lower level when long-range transported airmass did not have a big influence in London. On the contrary, the enhanced SO$_4$ concentration during lockdown spring is due to the airmass is originated from the sea (e.g., MSA) or long-range transported continental airmasses.

[Figure]

5. Line 222: Please refer to which figure describes long-range transport. In addition, I would like to see if SO4 or other pollutants are indeed related to long-range transport via CWT or PSCF backtrajectory analysis.

   The CWT of Org, NO3, SO4, NH4, and BC, which confirmed our statement that NH4, NO3, and SO4 are mainly originated from long-range transport. The CWT of Chl is not included as the non-refractory chloride in the atmosphere in Europe is often low (Bressi et al., 2021), therefore, the Chl measurement by ACSM is around detection limit with high uncertainties. And, yes, Figure S7 is added in the end of this sentence.

[Figure]

6. Line 229: Wouldn't the reduced BC concentration from winter to spring be related to less heating or energy consumption?

In central London, residential heating is mainly from natural gas, not wood burning. Also, BC in the measurement site is a roadside station, where vehicle exhaust emission is expected to be the main source of BC. The aethalometer source apportionment model for BC often results in noisy or negative solid fuel BC contribution in the measurement site even during wintertime. Therefore, we are confident the reduced BC concentration from pre-lockdown winter to pre-lockdown spring is mainly caused by the reducing vehicle mileages.

7. Line 241: I find it odd that the weekday comparison of BC concentration appears suddenly in this section. Since it is not a critical discussion point of the paper, I suggest removing it.

Thank you for the suggestion. It's an important message that the BC also went up with EOTHO from Mon-Wed with the increased traffic and cooking activities since the EOTHO was only valid from Mon to Wed (Aug 3rd–Aug 31st, 2020). To clarify this, the sentence has been revised as follows:

*"Since the EOTHO was only in place from Mon to Wed, BC concentrations (likely due to increased traffic and cooking emissions) increased on Mon-Wed compared to post-lockdown but before EOTHO (Jun 24th–Aug 2nd, 2020) (Figure S10)."*

8. Line 256: From my perspective, strong seasonality in OA factors is not as evident as the differences between the pre-COVID and lockdown periods. This is because the total OOA did not change significantly, and the PMF OOA separation carries more uncertainties compared to POA-related factors.

It is true that seasonality of OA sources not strong and here is clearly out-weighted by the effects of COVID-related policies. However, the seasonality of OA sources was still considerably obvious as shown in the figure below, which has been added as Figure S3.

[Figure]

*Figure S 1 Stacked mass concentration of OA sources for different seasons during the measurement period.*

The text was also amended to avoid confusion:

*"OA factors also showed considerable seasonality besides the effects from COVID-related policies (Figure 4 and Figure S3)."*

It is true OOA sources have larger uncertainties than POA factors in PMF analysis as they have larger fractions in OA as well as they are often not constrained with ME-2 during PMF analysis. However, the variabilities of OOA mass concentration and fractions are still non-negligible according to this previous Europe-wide study (including another urban background site in London, Chen et al., 2022). Therefore, the seasonality of OA factors was still briefly discussed in this manuscript.

9. Line 258: Would there be any relationship between the decrease in heating and energy consumption?

The decrease in spring compared to winter could partially be due to the decrease in heating and energy consumption, while the heating and energy consumption in central London are mainly from natural gas and renewable energy instead of solid fuel combustions. Therefore, the decrease in POA (HOA, COA, and BBOA) is not mainly caused by this reason. Here, we clarify the statement by adding following sentences:

*"It's worth mentioning that the reduced POA concentrations in warm season was not caused by reduced residential heating and energy consummation since Central*

*London mainly uses natural gas and renewable energy instead of solid fuel combustions."*

10. Line 261: Wouldn't photolysis affect the evaporation of some semi-volatile components in OOA? How about the f44 & f43 comparison by season?

Agreed, this is a typo from our side. I meant contributions. The sentence is revised as follows:

*"The OOA factor concentrations remain relatively consistent across seasons, while its contributions were larger during the warmer seasons. This is because both high temperature and strong irradiation will enhance the photochemistry and evaporation of POA sources and the increased biogenic volatile organic compound (VOC) emissions lead to high OOA production despite the evaporation of semi-volatile OOA."*

Also, the f44 vs f43 for winter and summer are provided below, where the seasonalises as well as positions of resolved OOAs are consistent with other studies across Europe (Canonaco et al., 2015; Chen et al., 2021, 2022).

[Figure]

11. Line 261: Where can I find increased VOC emissions in summer? Please provide any related references or data.

It should be increased biogenic VOC emissions in summer. The sentence is therefore revised:

*"The OOA factor concentrations remain relatively consistent across seasons, while its contributions were larger during the warmer seasons. This is because both high temperature and strong irradiation will enhance the photochemistry and evaporation of POA sources and the increased biogenic volatile organic compound (VOC) emissions lead to high OOA production despite the evaporation of semi-volatile OOA."*

12. Figure 2 right panel: I suggest having an independent axis for the OA data of the lockdown period.

We want to demonstrate the difference in diurnal cycles for all OA factors before, during and after lockdown instead of showing the diurnal variations of OA factor for each period. Therefore, we believe the same axis is important to convey this message. Thus, the right panel of Figure 2 is kept as it is.

13. Line 275-276: What does the smaller diurnal variation of MO-OOA compared to LO-OOA suggest? Could LO-OOA be associated with a specific source? Please provide a more detailed discussion.

MO-OOA across Europe in general shows smaller diurnal variation compared to LO-OOA as suggested in Fig.4 in Chen et al. (2022) shown below. Also, LO-OOA is also known as semi-volatile OOA (SV-OOA), which showed lower concentration during the daytime compared to the nighttime due to the effect of evaporation during the day and shallow boundary layer in the evening.

[Figure]

14. Line 279: If you zoom in, there could be a small COA peak during lunchtime. Please see Specific Comments #12.

Thanks a lot for pointing this out, we therefore changed the sentence accordingly:

*"The diurnal variation of COA and BBOA during lockdown showed much less intensity overall but the distinctive lunch peak remained as shown in the pre-lockdown; and the evening peak reduced its intensity (Figure 2)."*

15. Line 285-289: I believe this part does not fit well in the "Diurnal Cycle" section. Instead, it would be more appropriate in the next section, where the impact of the lockdown is discussed.

This part has been moved to the next section now as suggested.

16. Line 289: Figure S3: What was the backtrajectory like before the lockdown? I believe this information should be included to confirm whether the change in PM composition is indeed attributable to the lockdown policy.

The back trajectory plots for pre-lockdown spring and lockdown spring are shown below and has been added as Figure S5, where trajectories are coloured by dates. The high OOA concentration dates at the end of pre-lockdown spring and lockdown spring are clearly showed influences by continental airmass from Europe.

Pre-lockdown Spring          Lockdown Spring

17. Line 293-296: Does the data used to generate Figure 4 encompass the entire period? If so, I believe this information represents a key OA characteristic of central London and should be moved to the beginning of Section 3.2, rather than placed under the "Diurnal" section, to emphasize its significance. Additionally, making comparisons with other major cities, such as New York and Beijing, would be beneficial.

Yes, it includes the entire period and now has been moved up to the beginning of the section 3.2. The comparisons had been done in (Chen et al., 2022) as shown below, where London site is an urban background station measured from 2015-2018. However, comparisons with other major cities are out of the scope of this study.

[Figure]

Nevertheless, we revised the sentence to compare with other European cities and another source apportionment results in London urban background site as shown below:

*"This suggests that cooking emissions in Central London are responsible for elevated OA concentrations, which was also the case in Athens as shown in Chen et al. (2022)."*

18. Line 307: Please add a figure in SI that shows a temperature variation.

Added the following figure in SI as Figure S4.

[Figure]

19. Line 313: Please include any results that allow readers to determine whether photochemistry was increased in pre-lockdown spring compared to March 2020.

As shown in Figure S4 (point 18), the increased temperature and ozone level were evident with the uptick of the OOA fractions. Also, the back trajectory analysis proof that the long-range transportation of continental airmass was responsible for this increase. To be clearer, we revised the sentence as follows:

*"MO-OOA and LO-OOA increased by 136% and 279%, respectively due to long-range transportation of airmasses from continental Europe (Figure S5 and Figure S6) and increased photochemistry (enhanced temperature and ozone levels in Figure S4) compared to the first 25 days in Mar 2020."*

20. Section 3.2.4: I suggest relocating the POA part of this section to the front to emphasize the impact of EOTHO.

Agreed, moved to the end of this section now.

21. Line 341 & 345: The fact that EOTHO operated only from Monday to Wednesday may explain why COA did not return to pre-COVID levels. Therefore, Figure 6 should include a pre-COVID weekday plot and discuss if the difference in COA for the rest of the week would be responsible for the COA not being returned to pre-COVID levels.

Agreed, the figure 5 and figure 6 are merged into one figure to show the weekly cycles as well as shown below:

[Figure]

22. Line 342: From Figure 5, it is hard to recognize a 10% increase in COA from EOTHO to post EOTHO. Please show a figure that clearly shows such a difference or remove this statement.

True, this was a miscalculation from our side. This statement has been deleted as it's at the same level between EOTHO and post EOTHO.

23. Line 370: Which cities do you refer to? Please specify and add citations

This statement is true but not really important to this study, therefore, excluded from this sentence as follows:

*"Clearly detecting this change confirms the presence of COA (20% to OA) as an important source of OA in London and the importance of commercial cooking as a source."*

**Minor comments:**

1. Line 47: Please add references that show PM2.5 composition-dependent health effects and hospitalization:
   - Pye, H. O. T., Ward-Caviness, C. K., Murphy, B. N., Appel, K. W., and Seltzer, K. M.: Secondary organic aerosol association with cardiorespiratory disease mortality in the United States, Nature Communications, 12, 7215, 10.1038/s41467-021-27484-1, 2021.
   - Joo, T., Rogers, M. J., Soong, C., Hass-Mitchell, T., Heo, S., Bell, M. L., Ng, N. L., and Gentner, D. R.: Aged and Obscured Wildfire Smoke Associated with Downwind Health Risks, Environmental Science & Technology Letters, 11, 1340-1347, 10.1021/acs.estlett.4c00785, 2024.

Added

2. Line 49: Please add references that show source apportioned PM association with health effects owing to oxidative stresses:

- Vasilakopoulou, C. N., Matrali, A., Skyllakou, K., Georgopoulou, M., Aktypis, A., Florou, K., Kaltsonoudis, C., Siouti, E., Kostenidou, E., Błaziak, A., Nenes, A., Papagiannis, S., Eleftheriadis, K., Patoulias, D., Kioutsioukis, I., and Pandis, S. N.: Rapid transformation of wildfire emissions to harmful background aerosol, npj Climate and Atmospheric Science, 6, 218, 10.1038/s41612-023-00544-7, 2023.

- Liu, F., Joo, T., Ditto, J. C., Saavedra, M. G., Takeuchi, M., Boris, A. J., Yang, Y., Weber, R. J., Dillner, A. M., Gentner, D. R., and Ng, N. L.: Oxidized and Unsaturated: Key Organic Aerosol Traits Associated with Cellular Reactive Oxygen Species Production in the Southeastern United States, Environmental Science & Technology, 57, 14150-14161, 10.1021/acs.est.3c03641, 2023.

- Daellenbach, K. R., Uzu, G., Jiang, J., Cassagnes, L.-E., Leni, Z., Vlachou, A., Stefenelli, G., Canonaco, F., Weber, S., Segers, A., Kuenen, J. J. P., Schaap, M., Favez, O., Albinet, A., Aksoyoglu, S., Dommen, J., Baltensperger, U., Geiser, M., El Haddad, I., Jaffrezo, J.-L., and Prévôt, A. S. H.: Sources of particulate-matter air pollution and its oxidative potential in Europe, Nature, 587, 414-419, 10.1038/s41586-020-2902-8, 2020.

added

3. Line 56: Please add references related to PMF analysis on OA:

- Jimenez, J. L., Canagaratna, M. R., Donahue, N. M., Prevot, A. S. H., Zhang, Q., Kroll, J. H., DeCarlo, P. F., Allan, J. D., Coe, H., Ng, N. L., Aiken, A. C., Docherty, K. S., Ulbrich, I. M., Grieshop, A. P., Robinson, A. L., Duplissy, J., Smith, J. D., Wilson, K. R., Lanz, V. A., Hueglin, C., Sun, Y. L., Tian, J., Laaksonen, A., Raatikainen, T., Rautiainen, J., Vaattovaara, P., Ehn, M., Kulmala, M., Tomlinson, J. M., Collins, D. R., Cubison, M. J., Dunlea, J., Huffman, J. A., Onasch, T. B., Alfarra, M. R., Williams, P. I., Bower, K., Kondo, Y., Schneider, J., Drewnick, F., Borrmann, S., Weimer, S., Demerjian, K., Salcedo, D., Cottrell, L., Griffin, R., Takami, A., Miyoshi, T., Hatakeyama, S., Shimono, A., Sun, J. Y., Zhang, Y. M., Dzepina, K., Kimmel, J. R., Sueper, D., Jayne, J. T., Herndon, S. C., Trimborn, A. M., Williams, L. R., Wood, E. C., Middlebrook, A. M., Kolb, C. E., Baltensperger, U., and Worsnop, D. R.: Evolution of Organic Aerosols in the Atmosphere, Science, 326, 1525-1529, 10.1126/science.1180353, 2009.

- Zhang, Q., Jimenez, J. L., Canagaratna, M. R., Allan, J. D., Coe, H., Ulbrich, I., Alfarra, M. R., Takami, A., Middlebrook, A. M., Sun, Y. L., Dzepina, K., Dunlea, E., Docherty, K., DeCarlo, P. F., Salcedo, D., Onasch, T., Jayne, J. T., Miyoshi, T., Shimono, A., Hatakeyama, S., Takegawa, N., Kondo, Y., Schneider, J., Drewnick, F., Borrmann, S., Weimer, S., Demerjian, K., Williams, P., Bower, K., Bahreini, R., Cottrell, L., Griffin, R. J., Rautiainen, J., Sun, J. Y., Zhang, Y.

M., and Worsnop, D. R.: Ubiquity and dominance of oxygenated species in organic aerosols in anthropogenically-influenced Northern Hemisphere midlatitudes, Geophysical Research Letters, 34, https://doi.org/10.1029/2007GL029979, 2007.

added

4. Line 62 and 63: Please add references related to ACTRIS & ASCENT:

- Laj, P., Lund Myhre, C., Riffault, V., Amiridis, V., Fuchs, H., Eleftheriadis, K., Petäjä, T., Salameh, T., Kivekäs, N., Juurola, E., Saponaro, G., Philippin, S., Cornacchia, C., Alados Arboledas, L., Baars, H., Claude, A., De Mazière, M., Dils, B., Dufresne, M., Evangeliou, N., Favez, O., Fiebig, M., Haeffelin, M., Herrmann, H., Höhler, K., Illmann, N., Kreuter, A., Ludewig, E., Marinou, E., Möhler, O., Mona, L., Eder Murberg, L., Nicolae, D., Novelli, A., O'Connor, E., Ohneiser, K., Petracca Altieri, R. M., Picquet-Varrault, B., van Pinxteren, D., Pospichal, B., Putaud, J.-P., Reimann, S., Siomos, N., Stachlewska, I., Tillmann, R., Voudouri, K. A., Wandinger, U., Wiedensohler, A., Apituley, A., Comerón, A., Gysel-Beer, M., Mihalopoulos, N., Nikolova, N., Pietruczuk, A., Sauvage, S., Sciare, J., Skov, H., Svendby, T., Swietlicki, E., Tonev, D., Vaughan, G., Zdimal, V., Baltensperger, U., Doussin, J.-F., Kulmala, M., Pappalardo, G., Sorvari Sundet, S., and Vana, M.: Aerosol, Clouds and Trace Gases Research Infrastructure (ACTRIS): The European Research Infrastructure Supporting Atmospheric Science, Bulletin of the American Meteorological Society, 105, E1098-E1136, https://doi.org/10.1175/BAMS-D-23-0064.1, 2024.
- Hass-Mitchell, T., Joo, T., Rogers, M., Nault, B. A., Soong, C., Tran, M., Seo, M., Machesky, J. E., Canagaratna, M., Roscioli, J., Claflin, M. S., Lerner, B. M., Blomdahl, D. C., Misztal, P. K., Ng, N. L., Dillner, A. M., Bahreini, R., Russell, A., Krechmer, J. E., Lambe, A., and Gentner, D. R.: Increasing Contributions of Temperature-Dependent Oxygenated Organic Aerosol to Summertime Particulate Matter in New York City, ACS ES&T Air, 1, 113-128, 10.1021/acsestair.3c00037, 2024.
- Joo, T., Rogers, M. J., Soong, C., Hass-Mitchell, T., Heo, S., Bell, M. L., Ng, N. L., and Gentner, D. R.: Aged and Obscured Wildfire Smoke Associated with Downwind Health Risks, Environmental Science & Technology Letters, 11, 1340-1347, 10.1021/acs.estlett.4c00785, 2024.

added

5. Line 248: Please add "tracers" after "*m/z*".

added

6. Line 253: relocate "respectively" to Line 254, after "59% (26% to PM1)"

done

7. Line 299-302: I believe this should be combined into one sentence. Please revise.

Revised as follows:

*"OOA concentrations were relatively unaffected with some variability before, during, and after the lockdown since long-range transportation of airmasses from the continental Europe as observed for NH₄, NO₃, and SO₄."*

8. Line 301&312&320&333: Please refer to the figure that describes the long-range transport of different chemical species.

   Added

9. Line 305: Please add a reference (or add a figure in SI) about the GDP variation.

   Added a reference

10. Line 326-327: Please add references about the changes in travel, economic activities, and vehicle mileage during the measurement period.

    Added references accordingly.

11. Line 359-362: The statement is difficult to follow. Please revise.

    Revised as follows:

    *"PM concentrations increased at the beginning of the lockdown (Mar–Apr 2020) despite reduced economic activities, which was caused by long-range transported airmasses instead of primary emissions through examining the source apportionment (and inorganic PM composition)."*

**References**

Aksoyoglu, S., Jiang, J., Ciarelli, G., Baltensperger, U., & Prévôt, A. S. H. (2020). Role of ammonia in European air quality with changing land and ship emissions between 1990 and 2030. *Atmospheric Chemistry and Physics*, *20*(24), 15665–15680. https://doi.org/10.5194/acp-20-15665-2020

Bressi, M., Cavalli, F., Putaud, J. P., Fröhlich, R., Petit, J. E., Aas, W., Äijälä, M., Alastuey, A., Allan, J. D., Aurela, M., Berico, M., Bougiatioti, A., Bukowiecki, N., Canonaco, F., Crenn, V., Dusanter, S., Ehn, M., Elsasser, M., Flentje, H., … Prevot, A. S. H. (2021). A European aerosol phenomenology - 7: High-time resolution chemical characteristics of submicron particulate matter across Europe. *Atmospheric Environment: X*, *10*, 100108. https://doi.org/10.1016/J.AEAOA.2021.100108

Canonaco, F., Slowik, J. G., Baltensperger, U., & Prévôt, A. S. H. (2015). Seasonal differences in oxygenated organic aerosol composition: implications for emissions sources and factor analysis. *Atmospheric Chemistry and Physics*, *15*(12), 6993–7002. https://doi.org/10.5194/acp-15-6993-2015

Canonaco, F., Tobler, A., Chen, G., Sosedova, Y., Slowik, J. G., Bozzetti, C., Daellenbach, K. R., El Haddad, I., Crippa, M., Huang, R.-J., Furger, M., Baltensperger, U., & Prévôt, A. S. H. (2021). A new method for long-term source apportionment with time-dependent factor profiles and uncertainty assessment using SoFi Pro: application to 1 year of organic

aerosol data. *Atmospheric Measurement Techniques*, *14*(2), 923–943. https://doi.org/10.5194/amt-14-923-2021

Chen, G., Canonaco, F., Tobler, A., Aas, W., Alastuey, A., Allan, J., Atabakhsh, S., Aurela, M., Baltensperger, U., Bougiatioti, A., De Brito, J. F., Ceburnis, D., Chazeau, B., Chebaicheb, H., Daellenbach, K. R., Ehn, M., El Haddad, I., Eleftheriadis, K., Favez, O., … Prévôt, A. S. H. (2022). European aerosol phenomenology − 8: Harmonised source apportionment of organic aerosol using 22 Year-long ACSM/AMS datasets. *Environment International*, *166*, 107325. https://doi.org/10.1016/j.envint.2022.107325

Chen, G., Sosedova, Y., Canonaco, F., Fröhlich, R., Tobler, A., Vlachou, A., Daellenbach, K. R., Bozzetti, C., Hueglin, C., Graf, P., Baltensperger, U., Slowik, J. G., El Haddad, I., & Prévôt, A. S. H. (2021). Time-dependent source apportionment of submicron organic aerosol for a rural site in an alpine valley using a rolling positive matrix factorisation (PMF) window. *Atmospheric Chemistry and Physics*, *21*(19). https://doi.org/10.5194/acp-21-15081-2021

Simmonds, M., & George Finch. (2021). *The Impact of the Pandemic on UK Trade*. https://www.britishports.org.uk/content/uploads/2021/12/Pandemic-Impacts-on-UK-Trade-December-2021.pdf

---

## Author Comment (AC2)

**Response of the reviewers' comments on " *How COVID-19 related policies reshaped organic aerosol source contributions in central London"* by Gang I. Chen et al.**

*We thank all the constructive comments from two reviewers. The following texts are the response to the reviewers with normal italic font is original review comments, green font for authors' responses,* and the *blue italic font for changes in the revised version.*

*Review comments*

*Authors' response*

*Texts from the revised version*

**Reviewer #2**

Chen et al. present an original study on the impact of COVID-19 related social distancing policies in London, UK on the composition of atmospheric aerosols. Aerosol mass spectrometer measurements combined with source apportionment analysis allowed a dedicated focus on the different organic fractions. They highlight a sharp decrease in primary organic aerosols (eg traffic and biomass burning) during lockdown period (LD), but a significant increase of cooking organic aerosols during Eat Out To Help Out (EOTHO) policies.

In this study, the impacts of COVID-19 policies have been estimated by comparing LD and EOTHO periods to pre-periods, supposed to be representative of business as usual concentrations. After all the flourishing literature on this kind of study, I am rather concerned that the authors didn't take into account (nor at least discuss) the main limitation of this kind of analysis : the variability induced by meteorology. Did the authors check that their "business as usual" periods are representative, meteorologically speaking ? As examples, I don't think the pre-LD period is representative of the meteorological conditions during LD; conditons in June may also be different than in August. Precipitation, temperature, relative humidity, wind speed & direction, BLH (among others) can have a direct impact on the variability and the concentrations measured at a given site, and none of these are presented. This is especially dangerous when stating that component X has increased/decreased by Y% during lockdown (/EOTHO), because these results are very important for stakeholders, and can also be easily understood by non-expert citizens. To this regard, methodology is a critical aspect of the work, and must be as robust as possible. I am afraid that this is not quite the case here. As a main major revision, I suggest the authors to : investigate the meteorological representativeness of the different periods, and discuss the potential impacts on the results. If representativeness is not achieved, I suggest either to change methodology (by taking meteorology into account), or reshape the presentation of the results, by avoiding as much as possible to present numbered decrease or increase results.

We appreciate reviewer's positive comments on this manuscript. We also agree that the meteorological conditions can have significant impacts on the concentrations of PM/OA species. Therefore, we have conducted deweathering analysis using "worldmet" R package (Carslaw, 2025) to minimize the meteorological effects (i.e., RH, air temperature, wind speed, and wind direction). It generally it showed moderate effects for most of periods, except for the pre-lockdown spring period. More importantly, the effects of lockdown decreasing the primary organic aerosols (POA) and black carbon were even more evident for the deweathering analysis compared with the original dataset. Similarly, the EOTHO increased the COA by 88% instead of 100% after deweathering analysis. In addition, given that the data included business as usual seasons without lockdown and EOTHO policies, we have made figures below to isolate the effects from seasonality of the PM/OA species for both original and deweathered timeseries. Again, it demonstrated the evidence of how COVID-related policies had significant effects on both PM and POA mass concentrations as elaborated in the manuscript. Therefore, this study will present the original results by acknowledge the effects of meteorological effects via showing the deweathered results in SI.

[Figure]

[Figure]

Other major concerns :

- Introduction is not well structured. For instance, I don't know why the authors talk about aerosol mass spectrometers and source apportionment here, which is, in that case, more "material & methods" rather than an element of context justifying the interest of the study.

> We appreciate reviewer's comment on the structure of the introduction. This part has been moved to Methodology section.

- Section 2.4 is way too long. We don't need a general description of how ME-2 works, the authors need instead to provide all valuable information showing how the final PMF solution was obtained. Additionnally, BBOA has a rather constant contribution to OA throughout the different seasons (around 10-12%), even in summer. I am guessing that this may priorly come from the use of the rolling PMF rather than barbecue-ing or meat cooking. Plus, it is not clear if the authors constrained BBOA in their previous "summer" PMF. Did the authors also unambiguously find a solution with BBOA during summer only (with a profile ressembling to winter BBOA ?) ? The authors may also check BC data (and Angstrom exponent) to support the discussion.

> Agreed, general information about PMF, ME-2, etc have been removed now and more details in terms of how source apportionment was conducted have been added in this section.

> BBOA was also found in summer PMF in a seasonal PMF analysis in August 2019 as well as the summer 2023 dataset in the same site. It showed a similar diurnal variation as COA since there is evidence showing restaurants in the UK do use solid fuel to cook to meet customers' demands in flavours as a Defra's report suggests (Defra, 2023). With sufficient proof from the Defra report, we think additional analysis on BC data (i.e., Angstrom exponent) is not necessary to support our statement. In addition, in a roadside site, like Marylebone Road, the Aethalometer model to resolve BCwb is highly uncertain when BC is predominantly coming from traffic emissions.

Minor comments:

- Figure S3 (b) and (c) don't quite support long range transport. Maybe trajectory analysis (CWT, PSCF or cluster) would better help. Much more discussion is anyway needed concerning NO3, because it may not only arise from long range transport. Trajectory analysis may also help to look at the occurence of air masses (through eg cluster) throughout the different periods of the study. It would contribute to appreciate their meteorological representativeness.

The CWT of Org, NO3, SO4, NH4, and BC are provided in SI, which confirmed our statement that NH4, NO3, and SO4 are mainly originated from long-range transport. The CWT of Chl is not included as the non-refractory chloride in the atmosphere in Europe is often low (Bressi et al., 2021), therefore, the Cl measurement by ACSM is around detection limit with high uncertainties.

[Figure]

- Can the authors elaborate more on Figure S1, and especially how the different slopes obtained over time may impact the presented results. Is it an issue of ACSM calibration? or FIDAS measurements?

The ACSM RIE calibration factors during the measurement periods were averaged and then applied across the whole period, therefore, these discrepancies were not caused by calibrations, but rather different measurement techniques for FIDAS and ACSM. Therefore, the slope is often composition dependent. For instance, the low ACSM concentration but high FIDAS is caused by high fractions of refractory aerosols (e.g., sea salt) which cannot be measured effectively by ACSM. Also, the high ACSM+BC periods might be caused by the high fractions of ultra fine particles, which cannot be measured properly by the FIDAS. Therefore, the ACSM mass closure

practice is a sanity check with independent measurements rather than fully trust one instrument over the other.

- Comparison with literature and previous results elsewhere is clearly missing in almost all result sections.

In introduction, there is comprehensive discussions of general phenomena of the influences from COVID lockdown in previous studies. We do agree, there is little comparison with previous studies in the results section. Therefore, we have added some sentences in section 3.2.2 to mention the similarity and dissimilarity from current study and literature

**References**

Bressi, M., Cavalli, F., Putaud, J. P., Fröhlich, R., Petit, J. E., Aas, W., Äijälä, M., Alastuey, A., Allan, J. D., Aurela, M., Berico, M., Bougiatioti, A., Bukowiecki, N., Canonaco, F., Crenn, V., Dusanter, S., Ehn, M., Elsasser, M., Flentje, H., … Prevot, A. S. H. (2021). A European aerosol phenomenology - 7: High-time resolution chemical characteristics of submicron particulate matter across Europe. *Atmospheric Environment: X*, *10*, 100108. https://doi.org/10.1016/J.AEAOA.2021.100108

Carslaw, D. (2025). *deweather: Remove the influence of weather on air quality data*. Https://Openair-Project.Github.Io/Deweather/. https://openair-project.github.io/deweather/

Defra. (2023). *Use of domestic fuels in the hospitality sector – a qualitative review of hospitality businesses*. www.gov.uk/defra

---

## Author Response (AR2)

**Response of the reviewers' comments on "How COVID-19 related policies reshaped organic aerosol source contributions in Central London" by Gang I. Chen et al.**

We thank the two reviewers for all the constructive comments. The following response to the reviewers, details original review comments with normal italic font, green font for authors' responses, and blue italic font for changes in the revised version.

**Reviewer #1**

The authors addressed my comments well, and I now think this is acceptable for publication in Atmospheric Chemistry and Physics. I only have a few technical comments below. Line numbers refer to the cleaned, resubmitted version of the manuscript.

Thank you for the comments. We have addressed the comments as follows:

Line 199: Please revise "volatility of NH4NO3" to "semi-volatile nature of NH4NO3."

done

Line 201: Add "across Europe" after "... enhanced photochemistry."

done

Line 202: In addition to long-range transport, please note that airmasses originating from the sea can also affect SO4 enhancement, as mentioned in the author's response to comment #4.

Thank you for the comment, we have revised the sentence as shown below:

"During the lockdown in spring, SO4 concentrations remained high, which was associated with long-range transport and marine aerosols (e.g., methanesulfonic acid, MSA) (Fig. S7)."

Lines 247–248: Still, seasonality is not clear as the effect of COVID-related policies from my perspective, considering the uncertainties of Q-ACSM. I suggest toning down this statement by removing "considerable."

done

*Line 253-258: Please add appropriate references for this statement.*

Two citations have been added as suggested.

"It's worth mentioning that the reduced POA concentrations in the warm season was not caused by reduced residential heating and energy consummation since Central London mainly uses natural gas and renewable energy rather than solid fuel combustion (Cliff et al., 2025). The OOA factor concentrations remain relatively consistent across seasons, while its contribution was larger during the warmer seasons (Fig. S4). This is because both high temperature and strong solar radiation will enhance the photochemistry and evaporation of POA sources, and increased

volatile organic compound (VOC) emissions lead to high OOA production despite the evaporation of semi-volatile OOA (Fig. S4) (Chen et al., 2022)."

Line 275: Briefly include the point you made in response to comment #13 to support the small diurnal variations observed for MO-OOA and LO-OOA.

**Corrected as shown below:**

"The MO-OOA showed a smaller diurnal variation compared to LO-OOA. This is because the LO-OOA is also known as semi-volatile OOA (SV-OOA), which evaporate during the day due to higher temperatures and accumulate in the evening due to the shallower boundary layer (Chen et al., 2022)."

Line 320: How about adding a dedicated section on pre-EOTHO? I assume the authors separated this period because the lockdown policy was suspended, right? Since this period is also distinguished in Figure 4, it may be beneficial to treat it as its own section, but I'll leave it to the authors to decide.

Agreed, we have made it as a dedicated section as suggested. It now reads:

**"3.2.4 Pre-EOTHO**

During pre-EOTHO (Jun 24th–Aug 2nd, 2020) after the lockdown policy was eased, HOA, COA, and BBOA all showed considerably increases of 16%, 30%, and 14%, respectively when compared to lockdown summer period. In which, MO-OOA increased by 45% and LO-OOA decreased by 13%, respectively as the relatively higher temperature and solar radiation favoured the evaporation of LO-OOA and production of MO-OOA from LO-OOA and POA. As shown in Fig. 5, the POA concentrations were much lower when compared to summer 2019 (Aug 1st–Aug 31st, 2019) as travel and economic activities did not return to pre-COVID levels (ONS, 2022; Transport for London, 2020). Specifically, lower HOA (-33%), BC (-37%) due to reduced vehicle mileage resulted, and COA (-59%) due to the reduced commercial cooking activity. As BBOA is co-emitted with COA during of cooking activities, BBOA also decreased slightly from 0.53 to 0.38 μg/m3 (-28%, Fig. 5)."

**Reviewer #3**

This manuscript presents a year-long ACSM dataset to investigate the influence of short-term anthropogenic emission perturbations on the chemical composition of urban aerosols in London, with a particular focus on the source-resolved organic aerosol (OA) components. The dataset itself is valuable and potentially informative. From the content of the manuscript, this is a revised version addressing previous reviewer comments. However, it seems that the authors have not made a concerted effort to improve the manuscript to the standard required for publication in ACP. There are still numerous instances of careless editing errors. For example, on page 3, line 53: "Click or tap here to enter text.", and on page 12, line 232: "tError!".

We agree with the reviewer that this work is valuable and will be informative, especially for policymakers to improve air quality in urban environments. We apologise for the editorial errors, which have been all addressed in the revised version.

The writing also requires substantial improvement in terms of logic, coherence, rigour, and clarity. For instance, the Introduction is particularly difficult to follow. The narrative does not clearly articulate the main scientific question or its necessity in the context of prior studies. The figures also need improvement. Some of them, such as Figures 3 and 5, contain labels and text that are too small to be legible, and Figure 5 appears to be missing axis lines altogether.

We apologise the small mistakes we made in the last revision, which have been dealt with in this version.

Figure 1 Yearly averaged profiles (a) and diurnal cycles (b) of resolved factors from the rolling PMF analysis at the MY site.

Time is expressed in local time.

Figure 2 The impacts on OA sources during different periods compared with business-as-usual cases with and without deweathering analysis.

Also, the whole introduction has been rewritten as follows to more coherently outline the context of the study, previous research and the impact of the study:

**"1 Introduction**

Atmospheric particulate matter (PM) are tiny particles suspended in the air, which impact the climate directly and indirectly (IPCC, 2021; Seinfeld et al., 2006), and cause adverse human health effects (Kelly and Fussell, 2012; World Health Organization, 2021). The PM present in urban areas, such as London, is emitted directly or indirectly from a wide range of natural and anthropogenic sources, can be changed through atmospheric reactions and remain airborne for many days. It is consequently a complex mixture including inorganic species (metals, minerals, black carbon, nitrate, sulphate, etc.) and thousands of organic compounds whose origins remain too complicated to fully quantify. PM2.5 (PM with aerodynamic diameter smaller than 2.5 µm) is strongly associated with increased risks of cardiovascular and respiratory related mortalities and hospital admissions (Dominici et al., 2006; Joo et al., 2024; Pye et al., 2021; Wei et al., 2022, 2024). Some studies (Lippmann et al., 2013; UK Health Security Agency (UKHSA), 2022) have begun to demonstrate that some PM constituents and sources have stronger associations with a range of health metrics, including mortality, morbidity, and toxicities although the evidence remains inconsistent (Kelly and Fussell, 2012; Liu et al., 2023; Vasilakopoulou et al., 2023). With 99% of the urban population in Europe exposed to PM2.5 concentrations exceeding the WHO air quality guideline (European Environment Agency, 2024; World Health Organization, 2021), delivering clean air is a target for European and international governments according to the EU air quality directive (European Union, 2024). However, delivering publicly acceptable policies to improve air quality

remains challenging (Mebrahtu et al., 2023; Oltra et al., 2021). Targeting the sources of PM that are most health-relevant could be a more cost-effective (Wu et al., 2023), more easily communicated and more publicly acceptable approaches (Pinakidou, 2025) to improve public health. It is therefore important to better quantify the sources of PM and understand how they respond to policy interventions.

The COVID-19 pandemic is a natural experiment to assess the impact of policies which, while not aiming to reduce PM2.5 concentrations, significantly restricted social and economic activities and consequently reduced emissions. During the UK national lockdown, people were ordered to stay at home, and all non-essential businesses were closed, including pubs, cafes and restaurants from the end of Mar 26th, 2020. Nonessential shops were allowed to open from Jun 15th, and the first national lockdown came to an end on Jun 23rd, 2020. However, pubs, restaurants, and cafes were only allowed to open from July 4th, 2020. The UK recorded a 2.5% drop in Gross Domestic *Product (GDP)* in the first quarter of 2020, partly as people reduced their own activity prior to national lockdown. This accelerated to a 19.8% fall in GDP in April to June 2020 and household spending fell by over 20% over this period, the largest quarterly contraction on record, which was driven by falls in spending on restaurants, hotels, transport, and recreation (ONS, 2022). The UK Government Eat Out to Help Out (EOTHO) Scheme is examined specifically in this study as it influenced emissions from the commercial cooking sector. It was designed to help the hospitality industry recover from the financial impact of the national lockdown, offering a 50% meal discount up to a maximum of £10, which operated Mon to Wed, Aug  $3^{rd}$  to Aug  $31^{st}$ , 2020.

While the impact of these lockdown policies on some air quality metrics was smaller than expected given the large change in emissions (Shi et al., 2021), the abrupt nature of the intervention ensures it is easier to detect than other air quality policies that are more incremental in nature (Mudway et al., 2019). Some studies have investigated the lockdown impacts on chemical composition and sources of PM, which mainly focused on cities in China (Hu et al., 2022; Tian et al., 2021; Xu et al., 2020), a kerbside site in Toronto, Canada (Jeong et al., 2022), and an urban background site in Paris, France (Petit et al., 2021). These studies all resolved primary sources including traffic related emissions, biomass burning emissions from residential heating, cooking emissions (except Paris), and secondary sources from PMF analysis on organic aerosol (OA). Traffic and cooking emissions appeared to decrease during the lockdown in all sites, while biomass burning predominately from residential heating sources in Chinese cities increased as result of remote work and rather early lockdown measures (Jan-Feb 2020) compared to France. Secondary organic aerosol (SOA) showed a more complex phenomenon given its abundance in organic components and dynamic spatiotemporal conditions. Overall, the lockdowns resulted in decreased SOA in both northwest cities in China (Tian et al., 2021; Xu et al., 2020) and Paris (Petit et al., 2021) due to lower primary emissions, and therefore fewer SOA formation products. However, Beijing experienced a large increase in SOA concentrations due to increased fossil fuel and biomass burning emissions, long-range transport influences as well as favourable meteorological conditions (high RH, low wind speed and low boundary layer height) for SOA formation during the lockdown period (Hu et al., 2022). Therefore, the lockdown effects on the SOA were dependent on the abundance of primary emissions, long-range transported air masses, and meteorological conditions. To date, there are few studies that investigate how COVIDrelated policies could have impacted PM chemical composition and sources. Petit et al. (2021) and Gamelas et al. (2023) are the only two studies in Europe. The unique

COVID-related policies in the UK therefore provide a rare opportunity to investigate the impacts these policies had on chemical composition and OA. We used highly time resolved measurements from an air quality supersite located in the Central London from 2019 to 2020, and advanced source apportionment approaches to quantify the PM composition and OA sources before, during and after the UK national lockdown and EOTHO scheme. This study provides valuable insight into PM sources and composition in a global mega city and how air quality responds to abrupt changes in emissions from different sources. Importantly, it helps to establish the importance of cooking as a source of PM and uniquely associates biomass burning organic aerosol with commercial cooking emissions. This provides crucial information to policy makers as they attempt to reduce exposure to air pollution in urban areas."

In this revised version, the authors have included the results from deweathering analyses. However, no details are provided regarding the modelling methodology, model configurations, validation approach, and/or cross-validation results. Such critical information is essential and should be explicitly described. Additionally, the meteorological parameters considered (i.e., relative humidity, wind speed, wind direction, and temperature) are rather limited and insufficient to represent the complexity of meteorological influences on atmospheric aerosol variability. It is strongly recommended that the authors include other relevant meteorological indicators such as planetary boundary layer height and air mass trajectory cluster analysis. Relevant references that may help improve this section include Grange and Carslaw (2019, Sci. Total Environ.) and Shi et al. (2021, Sci. Adv.).

Thank you for reviewer's comment on the deweathering analyses in our work. We have now included additional detail of the modelling methodology, model configurations, validation approach, and/or cross-validation results.

Regarding the inclusion of additional meteorological indicators other authors (e.g. Grange and Carslaw, (2019)) have suggested boundary layer height, air mass cluster, or back trajectory information would be beneficial to include to deal with pollutants primarily controlled by regional scale process. However, the aim of this study is understanding how COVID-related policies affect primary/local emission sources (i.e., BC, HOA, COA, and BBOA), which will not be affected significantly by regional processes, and we do not feel that the additional metrics will improve the quantification of these PM components. It is consistent and comparable with previous studies (Font et al., 2022; Grange et al., 2021; Yao and Zhang, 2024) that includes similar meteorological parameters (i.e., wind speed, wind direction, relative humidity, temperature). It has already achieved excellent model performances with an R2 larger than 0.77, which are comparable with previous studies (Font et al., 2022; Grange et al., 2018, 2021; Grange and Carslaw, 2019; Krechmer et al., 2018; Shi et al., 2021; Yao and Zhang, 2024) All of which have provided scientifically insightful findings. Thus, the overall conclusion will not be changed even with the new analysis. To clarify this, we have added necessary text in the main text (Section 2.5 and 3.1).

Table S 1 Boot regression trees model performance on the testing dataset for each PM species/sources

|                 | Slope | R² (Pearson) | RMSE |
|-----------------|-------|--------------|------|
| ВС              | 0.98  | 0.94         | 0.46 |
| Org             | 1.01  | 0.93         | 1.80 |
| NH 4 | 1     | 0.87         | 0.39 |
| NO 3 | 1.02  | 0.92         | 1.52 |
| SO ₄     | 1.01  | 0.94         | 0.63 |
| HOA             | 0.97  | 0.77         | 0.34 |
| COA             | 0.99  | 0.82         | 0.66 |
| BBOA            | 1     | 0.86         | 0.29 |
| MO-OOA          | 1.01  | 0.92         | 0.64 |
| LO-OOA          | 1.01  | 0.89         | 0.45 |

To expand on model performance, we have now included the performance of our BRT model for each PM species/source in Table S2 in SI as shown above. The performance are good to excellent with slopes from 0.97 to 1.02 and R2 (Pearson) from 0.77 to 0.94, which are comparable and consistent with previous studies (Font et al., 2022; Grange et al., 2018, 2021; Grange and Carslaw, 2019; Krechmer et al., 2018; Shi et al., 2021; Yao and Zhang, 2024). As suggested by Grange et al. (2018) and Yao and Zhang, (2024), BRT models generally suffer from overfitting, however, this does not appear to be significant in our study (i.e., generally consistent good performances for both training and testing datasets). This is in contrast to random forest models, which normally provide poorer statistical agreement (Grange et al., 2018; Yao and Zhang, 2024). However, as our BRT model already provides good to excellent predictions without overfitting, we do not consider it necessary to perform random forest model by including additional meteorological parameters. Previous authors have also demonstrated that the BRT and random forest models show generally similar results (Yao and Zhang, 2024). Therefore, as the scope of this study is not deweathering and different models with different parameters will not change the results significantly, especially it will not change our conclusions how COVID-related policies will affect the primary emissions, we have decided not to perform the additional analyses reviewer suggested. However, to clarify the model performance, rationale for model and parameter selection, an additional section has been added in the Methodology section (Section 2.5 and 3.1) as follows:

**"2.5 Meteorological normalisation using boot regression tree model**

Meteorological normalisation, also known as deweathering analysis, has been conducted using the "worldmet" R package (Carslaw, 2025) to build boot regression tree (BRT) models for all resolved OA factors from PMF as well as chemical species measured. Considered variables, included relative humidity, wind speed, wind direction, and air temperature trend, hours of the day (local time), day of the week, Julian dates, week of the year as suggested by (Carslaw, (2025). While Grange and Carslaw (2019) have also suggested boundary layer height, air mass cluster, or back trajectory information would be beneficial to include to deal with pollutants primarily controlled by regional scale process. However, the aim of this study is understanding how COVID-related policies affect primary/local emission sources (i.e., BC, HOA, COA, and BBOA), which will not be affected significantly regional process, therefore additional metrics will most likely not improve the quantification of these PM

components. It is consistent and comparable with previous studies (Font et al., 2022; Grange et al., 2021; Yao and Zhang, 2024) that includes similar meteorological parameters (i.e., wind speed, wind direction, relative humidity, temperature). In addition, since the trained BRT models are sufficiently good even without considering boundary layer height and back trajectories, performing random forest model will not improve the model significantly, nor change the results drastically as suggested by Yao and Zhang (2024). Thus, in this study, only BRT models were trained and the meteorological effects subsequently removed (i.e., relative humidity, wind speed, wind direction, and air temperature) on all PM/OA.

**3 Results and Discussions**

**3.1 Model performance of meteorological normalisation**

The performance of each model (for individual species/source) is shown in Table S2 with slopes from 0.97 to 1.02 and R² (Pearson) from 0.77 to 0.94, which have similar or somewhat better performances compared with previous studies (Font et al., 2022; Grange et al., 2018, 2021; Grange and Carslaw, 2019; Krechmer et al., 2018; Shi et al., 2021; Yao and Zhang, 2024). As shown in the SI (Fig. S8 and Fig. S9) and the lower panel of Fig. 5, meteorological effects were generally considerable, especially for Pre-lockdown Spring period, while it does not change the conclusion of the effects from lockdown and EOTHO policies. Therefore, the main results presented in this study are based on the original measurements."

Overall, the work may be more appropriately considered as a measurement-report type submission. Major revisions are necessary before the manuscript can be considered suitable for publication in ACP.

Thank you for the comment, however, we strongly disagree with the reviewer on this point. This work has potentially high impact for policy makers internationally, it provides clear evidence of cooking as a major source on PM2.5 in urban areas, which is currently an under-recognised source, and co-emission of BBOA which has previously been wholly ascribed to wood burning. It therefore has the potential to provide important scientific justification for further studies and mitigation approaches. Such mitigation has been suggested in London government guidance but not yet acted upon. This is especially important as cities such as London have substantially reduced traffic emissions yet still do not meet international air quality targets.

Furthermore, the scope of this work fits perfectly with the special issue we have submitted to (<a href="https://acp.copernicus.org/articles/special\_issue1175.html">https://acp.copernicus.org/articles/special\_issue1175.html</a>), which specifically focus on following aspects:

- quantifying the spatial and temporal extent of stay-at-home policies on the European atmosphere, at both local and regional scales,
- evaluating the impact of lockdown measures on the formation of secondary pollutants,
- documenting the impact of reduced emissions (including air-traffic emissions) on cloud properties and occurrence, and
- estimating the "missing" emissions using observation—model approaches. The outcome from the special issue aims to provide an in-depth analysis of the

perturbation induced by the repeated lockdowns on the complex atmospheric system.

This work is therefore not only scientific significant, aligns with the scope of this special issue of the ACP, but is also impactful for policies both locally and worldwide.

In terms of writing, specific sentence improvements could be made. For instance:

1. Line 17 sentence could be revised to: "Organic aerosol (OA), a major component of submicron particulate matter (PM1), has significant impacts on both human health and climate. Quantifying its sources is therefore crucial for developing effective mitigation strategies."

**Revised**

2. Lines 19 sentence could be rewritten as: "Positive matrix factorisation (PMF) applied to aerosol chemical speciation monitor (ACSM) mass spectral data offers a robust approach for quantifying OA sources."

**Revised**

**References**

- Chen, G., Canonaco, F., Tobler, A., Aas, W., Alastuey, A., Allan, J., Atabakhsh, S., Aurela, M., Baltensperger, U., Bougiatioti, A., De Brito, J. F., Ceburnis, D., Chazeau, B., Chebaicheb, H., Daellenbach, K. R., Ehn, M., El Haddad, I., Eleftheriadis, K., Favez, O., Flentje, H., Font, A., Fossum, K., Freney, E., Gini, M., Green, D. C., Heikkinen, L., Herrmann, H., Kalogridis, A.-C., Keernik, H., Lhotka, R., Lin, C., Lunder, C., Maasikmets, M., Manousakas, M. I., Marchand, N., Marin, C., Marmureanu, L., Mihalopoulos, N., Močnik, G., Nęcki, J., O'Dowd, C., Ovadnevaite, J., Peter, T., Petit, J.-E., Pikridas, M., Matthew Platt, S., Pokorná, P., Poulain, L., Priestman, M., Riffault, V., Rinaldi, M., Różański, K., Schwarz, J., Sciare, J., Simon, L., Skiba, A., Slowik, J. G., Sosedova, Y., Stavroulas, I., Styszko, K., Teinemaa, E., Timonen, H., Tremper, A., Vasilescu, J., Via, M., Vodička, P., Wiedensohler, A., Zografou, O., Cruz Minguillón, M., and Prévôt, A. S. H.: European aerosol phenomenology 8: Harmonised source apportionment of organic aerosol using 22 Year-long ACSM/AMS datasets, Environ Int, 166, 107325, https://doi.org/10.1016/j.envint.2022.107325, 2022.
- Cliff, S. J., Drysdale, W., Lewis, A. C., Møller, S. J., Helfter, C., Metzger, S., Liddard, R., Nemitz, E., Barlow, J. F., and Lee, J. D.: Evidence of Heating-Dominated Urban NOx Emissions, Environ Sci Technol, 59, 4399–4408, https://doi.org/10.1021/acs.est.4c13276, 2025.
- Dominici, F., Peng, R. D., Bell, M. L., Pham, L., McDermott, A., Zeger, S. L., and Samet, J. M.: Fine Particulate Air Pollution and Hospital Admission for Cardiovascular and Respiratory Diseases, JAMA, 295, 1127, https://doi.org/10.1001/jama.295.10.1127, 2006.

- European Environment Agency: Europe's air quality status 2024, https://doi.org/10.2800/5970, 2024.
- European Union: Directive (EU) 2024/2881 of the European Parliament and of the Council of 23 October 2024 on ambient air quality and cleaner air for Europe (recast), 2024.
- Font, A., Ciupek, K., Butterfield, D., and Fuller, G. W.: Long-term trends in particulate matter from wood burning in the United Kingdom: Dependence on weather and social factors, Environmental Pollution, 314, 120105, https://doi.org/10.1016/j.envpol.2022.120105, 2022.
- Gamelas, C. A., Canha, N., Vicente, A., Silva, A., Borges, S., Alves, C., Kertesz, Z., and Almeida, S. M.: Source apportionment of PM2.5 before and after COVID-19 lockdown in an urban-industrial area of the Lisbon metropolitan area, Portugal, Urban Clim, 49, 101446, https://doi.org/10.1016/j.uclim.2023.101446, 2023.
- Grange, S. K. and Carslaw, D. C.: Using meteorological normalisation to detect interventions in air quality time series, Science of The Total Environment, 653, 578–588, https://doi.org/10.1016/j.scitotenv.2018.10.344, 2019.
- Grange, S. K., Carslaw, D. C., Lewis, A. C., Boleti, E., and Hueglin, C.: Random forest meteorological normalisation models for Swiss PM 10 trend analysis, Atmos Chem Phys, 18, 6223–6239, https://doi.org/10.5194/acp-18-6223-2018, 2018.
- Grange, S. K., Lee, J. D., Drysdale, W. S., Lewis, A. C., Hueglin, C., Emmenegger, L., and Carslaw, D. C.: COVID-19 lockdowns highlight a risk of increasing ozone pollution in European urban areas, Atmos Chem Phys, 21, 4169–4185, https://doi.org/10.5194/acp-21-4169-2021, 2021.
- Hu, R., Wang, S., Zheng, H., Zhao, B., Liang, C., Chang, X., Jiang, Y., Yin, R., Jiang, J., and Hao, J.: Variations and Sources of Organic Aerosol in Winter Beijing under Markedly Reduced Anthropogenic Activities During COVID-2019, Environ Sci Technol, 56, 6956–6967, https://doi.org/10.1021/acs.est.1c05125, 2022.
- IPCC: Climate Change 2021 The Physical Science Basis, Cambridge University Press, https://doi.org/10.1017/9781009157896, 2021.
- Jeong, C.-H., Yousif, M., and Evans, G. J.: Impact of the COVID-19 lockdown on the chemical composition and sources of urban PM2.5, Environmental Pollution, 292, 118417, https://doi.org/10.1016/j.envpol.2021.118417, 2022.
- Joo, T., Rogers, M. J., Soong, C., Hass-Mitchell, T., Heo, S., Bell, M. L., Ng, N. L., and Gentner, D. R.: Aged and Obscured Wildfire Smoke Associated with Downwind Health Risks, Environ Sci Technol Lett, 11, 1340–1347, https://doi.org/10.1021/acs.estlett.4c00785, 2024.
- Kelly, F. J. and Fussell, J. C.: Size, source and chemical composition as determinants of toxicity attributable to ambient particulate matter, Atmos Environ, 60, 504–526, https://doi.org/10.1016/j.atmosenv.2012.06.039, 2012.

- Krechmer, J., Lopez-Hilfiker, F., Koss, A., Hutterli, M., Stoermer, C., Deming, B., Kimmel, J., Warneke, C., Holzinger, R., Jayne, J., Worsnop, D., Fuhrer, K., Gonin, M., and de Gouw, J.: Evaluation of a New Reagent-Ion Source and Focusing Ion–Molecule Reactor for Use in Proton-Transfer-Reaction Mass Spectrometry, Anal Chem, 90, 12011–12018, https://doi.org/10.1021/acs.analchem.8b02641, 2018.
- Lippmann, M., Chen, L. C., Gordon, T., Ito, K., and Thurston, G. D.: National Particle Component Toxicity (NPACT) Initiative: Integrated Epidemiologic and Toxicologic Studies of the Health Effects of Particulate Matter Components, Boston, 2013.
- Liu, F., Joo, T., Ditto, J. C., Saavedra, M. G., Takeuchi, M., Boris, A. J., Yang, Y., Weber, R. J., Dillner, A. M., Gentner, D. R., and Ng, N. L.: Oxidized and Unsaturated: Key Organic Aerosol Traits Associated with Cellular Reactive Oxygen Species Production in the Southeastern United States, Environ Sci Technol, 57, 14150–14161, https://doi.org/10.1021/acs.est.3c03641, 2023.
- Mebrahtu, T. F., McEachan, R. R. C., Yang, T. C., Crossley, K., Rashid, R., Hossain, R., Vaja, I., and Bryant, M.: Differences in public's perception of air quality and acceptability of a clean air zone: A mixed-methods cross sectional study, J Transp Health, 31, 101654, https://doi.org/10.1016/J.JTH.2023.101654, 2023.
- Mudway, I. S., Dundas, I., Wood, H. E., Marlin, N., Jamaludin, J. B., Bremner, S. A., Cross, L., Grieve, A., Nanzer, A., Barratt, B. M., Beevers, S., Dajnak, D., Fuller, G. W., Font, A., Colligan, G., Sheikh, A., Walton, R., Grigg, J., Kelly, F. J., Lee, T. H., and Griffiths, C. J.: Impact of London's low emission zone on air quality and children's respiratory health: a sequential annual cross-sectional study, Lancet Public Health, 4, e28–e40, https://doi.org/10.1016/S2468-2667(18)30202-0, 2019.
- Oltra, C., Sala, R., López-asensio, S., Germán, S., and Boso, À.: Individual-Level Determinants of the Public Acceptance of Policy Measures to Improve Urban Air Quality: The Case of the Barcelona Low Emission Zone, Sustainability 2021, Vol. 13, Page 1168, 13, 1168, https://doi.org/10.3390/SU13031168, 2021.
- ONS: GDP and events in history: how the COVID-19 pandemic shocked the UK economy, 2022.
- Petit, J.-E., Dupont, J.-C., Favez, O., Gros, V., Zhang, Y., Sciare, J., Simon, L., Truong, F., Bonnaire, N., Amodeo, T., Vautard, R., and Haeffelin, M.: Response of atmospheric composition to COVID-19 lockdown measures during spring in the Paris region (France), Atmos Chem Phys, 21, 17167–17183, https://doi.org/10.5194/acp-21-17167-2021, 2021.
- Pinakidou, S.: People's perceptions of air pollution and their awareness of official indexes at the start of the twenty-first century: a review, Discover Environment 2025 3:1, 3, 1–20, https://doi.org/10.1007/S44274-025-00213-X, 2025.

- Pye, H. O. T., Ward-Caviness, C. K., Murphy, B. N., Appel, K. W., and Seltzer, K. M.: Secondary organic aerosol association with cardiorespiratory disease mortality in the United States, Nat Commun, 12, 7215, https://doi.org/10.1038/s41467-021-27484-1, 2021.
- Seinfeld, J. H., Wiley, J., and Pandis, S. N.: ATMOSPHERIC From Air Pollution to Climate Change SECOND EDITION, 628–674 pp., 2006.
- Shi, Z., Song, C., Liu, B., Lu, G., Xu, J., Van Vu, T., Elliott, R. J. R., Li, W., Bloss, W. J., and Harrison, R. M.: Abrupt but smaller than expected changes in surface air quality attributable to COVID-19 lockdowns, Sci Adv, 7, 6696–6709, https://doi.org/10.1126/SCIADV.ABD6696/SUPPL\_FILE/ABD6696\_SM.PDF, 2021.
- Tian, J., Wang, Q., Zhang, Y., Yan, M., Liu, H., Zhang, N., Ran, W., and Cao, J.: Impacts of primary emissions and secondary aerosol formation on air pollution in an urban area of China during the COVID-19 lockdown, Environ Int, 150, 106426, https://doi.org/10.1016/j.envint.2021.106426, 2021.
- UK Health Security Agency (UKHSA): Statement on the differential toxicity of particulate matter according to source or constituents, https://www.gov.uk/government/publications/particulate-air-pollution-health-effects-of-exposure/statement-on-the-differential-toxicity-of-particulate-matter-according-to-source-or-constituents-2022, 27 July 2022.
- Vasilakopoulou, C. N., Matrali, A., Skyllakou, K., Georgopoulou, M., Aktypis, A., Florou, K., Kaltsonoudis, C., Siouti, E., Kostenidou, E., Błaziak, A., Nenes, A., Papagiannis, S., Eleftheriadis, K., Patoulias, D., Kioutsioukis, I., and Pandis, S. N.: Rapid transformation of wildfire emissions to harmful background aerosol, NPJ Clim Atmos Sci, 6, 218, https://doi.org/10.1038/s41612-023-00544-7, 2023.
- Wei, Y., Qiu, X., Yazdi, M. D., Shtein, A., Shi, L., Yang, J., Peralta, A. A., Coull, B. A., and Schwartz, J. D.: The Impact of Exposure Measurement Error on the Estimated Concentration—Response Relationship between Long-Term Exposure to PM2.5 and Mortality, Environ Health Perspect, 130, https://doi.org/10.1289/EHP10389, 2022.
- Wei, Y., Feng, Y., Danesh Yazdi, M., Yin, K., Castro, E., Shtein, A., Qiu, X., Peralta, A. A., Coull, B. A., Dominici, F., and Schwartz, J. D.: Exposure-response associations between chronic exposure to fine particulate matter and risks of hospital admission for major cardiovascular diseases: population based cohort study, BMJ, e076939, https://doi.org/10.1136/bmj-2023-076939, 2024.
- World Health Organization: WHO global air quality guidelines. Particulate matter (PM2.5 and PM10), ozone, nitrogen dioxide, sulfur dioxide and carbon monoxide., Geneva, 2021.
- Wu, D., Zheng, H., Li, Q., Wang, S., Zhao, B., Jin, L., Lyu, R., Li, S., Liu, Y., Chen, X., Zhang, F., Wu, Q., Liu, T., Jiang, J., Wang, L., Li, X., Chen, J., and Hao, J.: Achieving health-oriented air pollution control requires integrating unequal toxicities of industrial

- particles, Nature Communications 2023 14:1, 14, 1–12, https://doi.org/10.1038/s41467-023-42089-6, 2023.
- Xu, J., Ge, X., Zhang, X., Zhao, W., Zhang, R., and Zhang, Y.: COVID-19 Impact on the Concentration and Composition of Submicron Particulate Matter in a Typical City of Northwest China, Geophys Res Lett, 47, https://doi.org/10.1029/2020GL089035, 2020.
- Yao, X. and Zhang, L.: Identifying decadal trends in deweathered concentrations of criteria air pollutants in Canadian urban atmospheres with machine learning approaches, Atmos Chem Phys, 24, 7773–7791, https://doi.org/10.5194/acp-24-7773-2024, 2024.